# Bayesian inference of RNA velocity incorporating timepoints, lineage bifurcations, and count data

Yichen Gu[1], Yuxuan Song[2], David Blaauw[1,3], Joshua D. Welch[3,2]*

**1** Department of Electrical and Computer Engineering, University of Michigan, Ann Arbor, Michigan, United States of America, **2** Department of Computational Medicine and Bioinformatics, University of Michigan, Ann Arbor, Michigan, United States of America, **3** Department of Computer Science and Engineering, University of Michigan, Ann Arbor, Michigan, United States of America

* welchjd@umich.edu

## Abstract

Experimental approaches for measuring single-cell gene expression can observe each cell at only one time point, requiring computational approaches for reconstructing the dynamics of gene expression during cell fate transitions. RNA velocity is a promising computational approach for this problem, but existing inference methods fail to capture key aspects of real data, limiting their utility. To address these limitations, we developed VeloVAE, a Bayesian model for RNA velocity inference. VeloVAE uses variational Bayesian inference to estimate the posterior distribution of latent time, latent cell state, and kinetic rate parameters for each cell. Our approach can incorporate prior distributions on rate parameters and time points; model lineage bifurcations using branching differential equations; and directly model discrete count data. We show that VeloVAE significantly outperforms previous approaches in terms of data fit, accuracy of inferred differentiation directions, and transcription rate estimation. These improvements allow VeloVAE to accurately model gene expression dynamics in complex biological systems, including hematopoiesis, induced pluripotent stem cell reprogramming, the developing mouse brain, and the entire mouse embryo. We find that the latent time automatically inferred using all cells can even outperform pseudotime inferred using manually chosen cell subsets and root cells. Our work provides important new tools for modeling sequential changes in gene expression from single-cell expression data.

## Author summary

Understanding how cells change their identity over time is a central question in biology. However, most experiments that measure gene activity in individual cells can only capture a single snapshot of each cell, making it difficult to see how cells move through developmental or disease-related processes. In our work, we developed a computational method VeloVAE to reconstruct these cellular

**Data availability statement:** Our code is available at https://github.com/welch-lab/VeloVAE. We also provide the reproducible notebooks and scripts for all the benchmark methods and datasets. All source code, including the implementations and hyperparameter configurations for each benchmark method, is also available at the github repository. For the datasets used in this paper, Pancreas, Dentate Gyrus, Mouse Erythroid and Bone marrow datasets could be obtained from scVelo api at https://scvelo.readthedocs.io/en/stable/api.html. The count matrix for the Mouse Hindbrain (pons) dataset is obtained from Kharchenko Lab at https://pklab.med.harvard.edu/ruslan/velocity/oligos/. The sequencing data for the mouse brain and Mouse Hindbrain (pons) datasets could be accessed in BioProject through PRJNA637987 and PRJNA637987. The sequencing data for the other datasets used in this study could be accessed through GEO: IPSC (GSE122662), Human Hematopoiesis (GSE102698), Human Erythroid (GSE167576), Mouse Retina (GSM3466902), Intestinal Organoid (GSE128365), Neurogenesis (GSE141851), Scifate2 (GSE236520), and Tracer (GSE220949).

**Funding:** This work was funded by NIH grant R01HG010883 to JDW. JDW and YG received salary from this NIH grant. The funders had no role in study design, data collection and analysis, decision to publish, or preparation of the manuscript.

**Competing interests:** The authors have declared that no competing interests exist.

trajectories from scRNA-seq data. Our approach uses a Bayesian framework to infer the latent time, cell state and the rates at which genes turn on or off. By integrating biological information and modeling complex processes such as cell type bifurcation, VeloVAE provides a more accurate and interpretable view of cell development than previous approaches. We demonstrate that VeloVAE can reveal key patterns of gene activity across several biological systems, including blood formation, brain development, and stem cell reprogramming. Overall, our work provides a general framework for understanding dynamic gene regulation from single-cell data.

## 1 Introduction

The human body contains many cell types with distinct forms and functions, which arise from progenitor cells in a sequential developmental process. A key question in molecular biology is what regulates this process of cellular development. Therefore, understanding cellular development requires modeling how mRNA expression changes over time. Such models are crucial for numerous areas of biology and medicine, such as neuroscience, cancer research, and regenerative stem-cell therapies.

Since its emergence, the single-cell RNA sequencing (scRNA-seq) technology [1] has been widely used to study cell development. However, scRNA-seq measurement destroys the cell, making it impossible to follow an individual cell longitudinally. Thus, computational approaches are required to assemble these static snapshots into a history of the gene expression changes occurring during a developmental process.

Two main types of computational approaches have been developed for this problem: pseudotime inference and RNA velocity. Pseudotime inference methods use distance from a manually-specified starting cell to rank cells according to degree of development [2,3]. Many pseudotime methods also aim to infer a tree or graph structure that represents the underlying structure of the developmental process [4–8]. In contrast, La Manno et al. [9] developed the concept of RNA velocity based on the observation that both unspliced and spliced mRNA molecules appear in sequencing outputs. The relative ratio of spliced and unspliced counts indicates whether the gene was being turned on or turned off at the time the cell was sequenced. La Manno et al. introduced an ODE model to describe the gene expression process, used a steady state assumption to estimate parameters, and implemented the method in a package called velocyto. Later work [10] relaxed the steady-state assumption, allowing all cells to be used in parameter estimation and inferring a latent time value for each cell. Bergen et al. implemented their method in a package called scVelo [10]. RNA velocity methods have been widely used by biologists to help understand cellular developmental processes [11–13]. With the advent of single-cell multiomics, multiple modalities bring a more comprehensive view of cell development. Consequently, the concept of RNA velocity is extended to different modalities, such as protein [14] and chromatin accessibility [15]. We recently extended the dynamical model of scVelo to incorporate chromatin accessibility data, packaged in a tool called MultiVelo [15].

While they have proven useful for biological discovery in many cases, existing approaches for pseudotime inference and RNA velocity inference have significant limitations. Pseudotime inference requires manually specifying a starting cell, is based purely on pairwise cell similarity, and cannot infer the directions or rates of cell development. RNA velocity addresses some of these limitations, and is in principle able to infer the directions, rates, and origins of developmental processes. However, current RNA velocity methods rely on numerous simplifying assumptions and fail to yield accurate results in many cases [16].

In particular, scVelo suffers from several significant limitations: (1) lack of a common time scale during model training (2) over-simplified ODE model with constant transcription rates (3) insufficient modeling capability for cell type bifurcations. To address these limitations, a number of methods have been developed. UniTVelo [17] improved time and velocity inference by assuming a unified time frame and more flexible parametric model. DeepVelo [18] and cellDancer [19] overcome the limitation of a simple ODE model by inferring cellwise transcription, splicing and degradation rates. Both methods exhibit better qualitative performance on more complex datasets. There is also on-going research in the field of Bayesian methods to solve the RNA velocity inference problem. VeloVI [20] applies a variational auto-encoder to jointly learn a posterior distribution of cell time and ODE parameters. PyroVelocity [21] applies Bayesian inference and directly models discrete counts instead of preprocessed continuous counts. While these methods improve the quality of RNA velocity estimation significantly, each of them has its own limitation. In some cases, UniTVelo [17] relies on the knowledge of the root cell to accurately reconstruct the dynamics, and its parametric model is not based on splicing kinetics, thus lacking a biochemical interpretation. DeepVelo [18] and cellDancer [19] need to estimate ground-truth velocity from a KNN graph, and don't infer cell time during model training. Cell time is computed post hoc and this leads to inconsistency between velocity and inferred time.

Due to these unresolved issues, it remains crucial to develop a more accurate computational method. In this work, we present VeloVAE, a Bayesian model that uses neural networks to jointly infer the posterior distribution of cell times, cell states, and gene expression rate parameters from scRNA-seq data. Our approach uses a simple, interpretable differential equation model to describe the dynamics of gene expression, but allows the parameters to vary continuously with cell state. We implemented the method as a variational auto-encoder, specifically a BasisVAE [22], to model the gene splicing dynamics as a mixture of two generative processes. The introduction of a cell state variable and a single latent time shared across all genes allows VeloVAE to model qualitative features of expression dynamics, including late induction, early repression, transcriptional boosts, and bifurcations. Consequently, VeloVAE can be used as a general tool to reconstruct the orders and rates of gene expression changes across many complex biological systems, including hematopoiesis, induced pluripotent stem cell reprogramming, neurogenesis, and organogenesis. Moreover, we connect RNA velocity to the chemical master equation (CME) by directly modeling discrete counts. Based on our framework, we deploy VeloVAE to generating discrete count data with underlying splicing kinetics described by ODEs. Our discrete VeloVAE model serves as a stepping stone towards solving a stochastic system described by the CME. In fact, the discrete VeloVAE model can be treated as a variational approximation to the solution of CME under certain assumptions (Methods). Relying on cell time and state inferred by VeloVAE, we further develop a new type of ODE model called Branching ODE (BrODE) that infers a cell-type relation graph and fits distinct kinetic rates for each cell type. This serves as a succinct yet more comprehensive ODE model to account for complexity in cell differentiation. We summarize the unique capabilities of the VeloVAE model compared to other RNA velocity models in Table 1.

## 2 Results

### 2.1 VeloVAE allows bayesian inference of cell times, cell states, and rate parameters

VeloVAE uses a Bayesian model for RNA velocity inference. We assume that we are given individual scRNA-seq profiles that measure the amounts of spliced ($s$) and unspliced ($u$) transcripts at single moments of a developmental process. Our goal is to use these observations to simultaneously infer the posterior distributions of underlying latent variables

**Table 1. Capabilities of RNA velocity models.**

| Method | Shared | Cell-wise | Bayesian | Raw counts | Gene-wise |
|---|---|---|---|---|---|
| | Latent time | ODE parameters | inference | modeling | bifurcation |
| scVelo | ✗ | ✗ | ✗ | ✗ | ✗ |
| UniTVelo | ✓ | ✗ | ✗ | ✗ | ✗ |
| DeepVelo | ✗ | ✓ | ✗ | ✗ | ✓ |
| cellDancer | ✗ | ✓ | ✗ | ✗ | ✓ |
| VeloVI | ✗ | ✗ | ✓ | ✗ | ✗ |
| PyroVelocity | ✓ | ✗ | ✓ | ✓ | ✗ |
| VeloVAE | ✓ | ✓ | ✓ | ✓ | ✓ |

that generated the data: cell time ($t$), cell state ($\mathbf{c}$), and rate parameters ($\theta$) describing the biochemical kinetics of gene expression. Our key modeling assumption is that the observed time-varying ($u(t),s(t)$) levels are related by a system of *two* ordinary differential equations (Fig 1A). As with previous RNA velocity approaches [9,10], these ODEs capture the simple insight that a gene must first be transcribed as nascent mRNA, then spliced into mature mRNA, and then subsequently degraded (Fig 1A). However, we make two important changes: (1) rather than assuming a single fixed transcription rate parameter for each gene across all cells, we allow each cell to have its own transcription rate $\rho$ for each gene; (2) gene counts are generated by a mixture of two ODE models. The first change removes the need for discrete induction and repression phases and models continuous changes in transcription rates, such as transcriptional boosts [16] and cell fate bifurcations. The second change considers two generative processes: (1) a gene is induced and may be repressed later; (2) a gene is repressed from the very beginning. Each gene has a probability of being generated by each of the two processes, which can be represented by a categorical random variable $w \sim p(w;\pi)$. This accounts for genes that only undergo repression and turns out to be important for the correctness of time reconstruction in certain dataset such as the erythroid lineage. Note that, unlike scVelo and VeloVI, which place each gene on a separate time scale, the cell time parameter $t$ is shared across all genes within a cell.

The VeloVAE model can be viewed from either an inference or a generative perspective (Fig 1B). From an inference perspective, if we know ($u(t),s(t)$) values for cells, we can infer something about the parameters, ($t,\mathbf{c},w,\pi,\theta$), that generated them (Fig 1B, left). Each particular location $\mathbf{c}$ in cell state space has an associated transcription rate $\rho$ for each gene; nearby cell state space locations will have similar transcription rates (Fig 1B, middle). From a generative perspective, if we know the ($t,\mathbf{c},w,\pi,\theta$) parameters for a cell, we can predict the distribution of their ($u,s$) values (Fig 1B, right). We can also incorporate prior information about the latent variables and rate parameters. If cell capture times are known (e.g., if cells were isolated separately on days 7 and 14), we can use the capture times as an informative prior for cell time.

To fit this statistical model on real data, we use autoencoding variational Bayes, a powerful statistical inference method in which neural networks approximate the posterior distribution of latent variables that may be nonlinearly related to observed data. Intuitively, autoencoding variational Bayes jointly trains the inference and generative models shown in Fig 1B, so that after training we can infer latent variables given observed data or predict new data given values of the latent variables. VeloVAE implements the inference model of Fig 1B using a neural network that takes ($u,s$) values as input and outputs the posterior distribution of cell time and cell state parameters (Fig 1C). VeloVAE implements the generative model of Fig 1B using a neural network that predicts the gene-specific transcription rates $\rho$ for each cell from the cell's time and state values (Fig 1C). The ($u,s$) values for each cell can then be predicted using the analytical solution to the ODE, which describes how spliced and unspliced counts vary over time (Fig 1C). We previously described the underlying theory and motivation of this approach in a machine learning conference paper [23]. Intuitively, VeloVAE is like an autoencoder whose decoder network has been replaced with the solution to a differential equation. Our previous work assumes

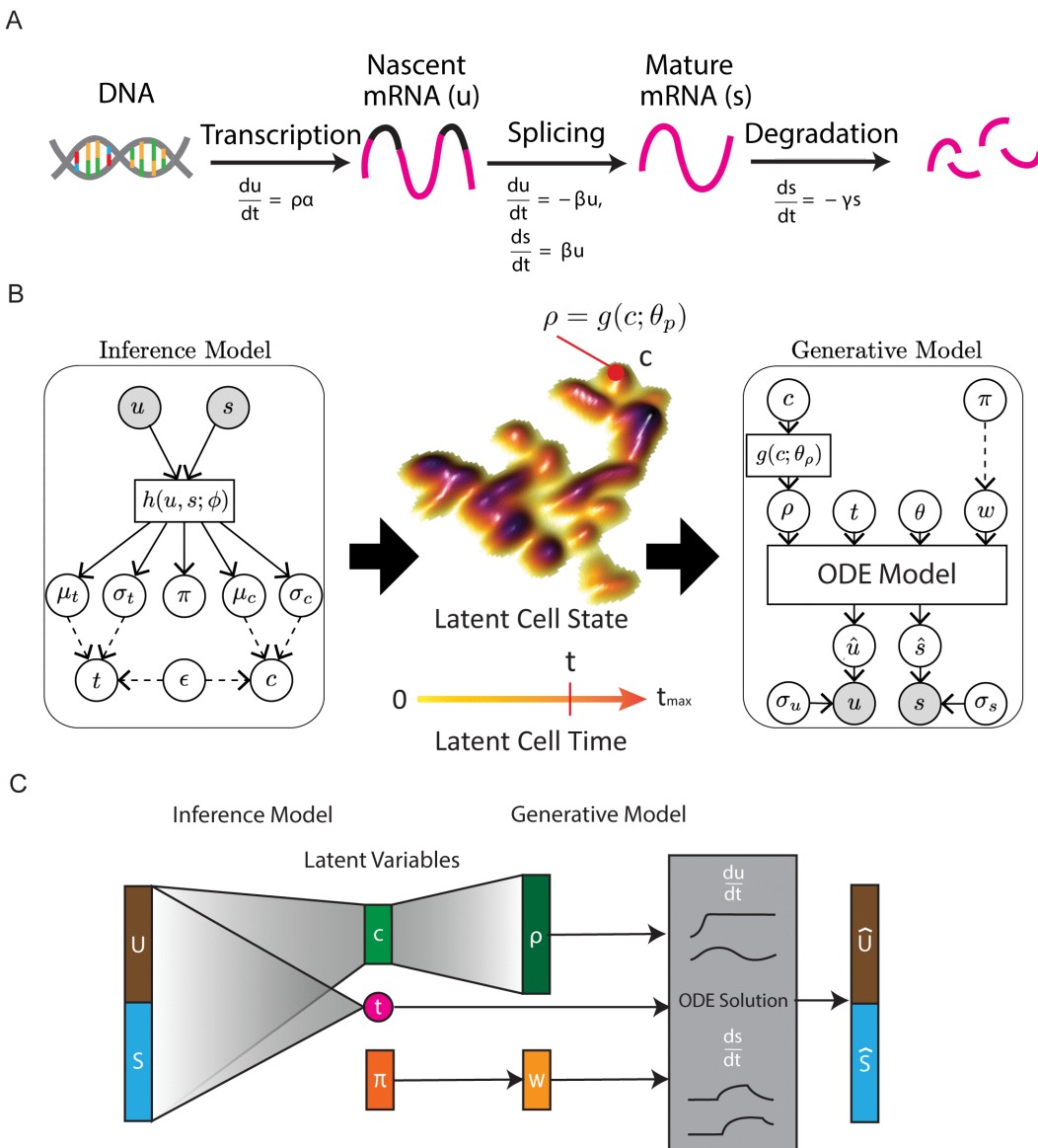

**Fig 1. VeloVAE model. (A)** Differential equation model of transcription used by VeloVAE. Nascent mRNA molecules are transcribed at gene- and cell-specific rate $\rho\alpha$. Next, nascent mRNA is spliced into mature mRNA at gene-specific rate $\beta$. Finally, mature mRNA is degraded at gene-specific rate $\gamma$. **(B)** Graphical model for latent variable inference and data generative processes. Cell time and state, as well as gene-wise mixture weight, are treated as latent variables, which are inferred using variational Bayes. Note that, unlike most previous approaches, latent time (and state) is shared across all genes within each cell. The latent cell state lies on a smooth low-dimensional manifold representing cell development. Each cell state has gene-specific transcription rates $\rho$. RNA count data are generated based on mixture of two ODE systems at the gene level with known analytical solutions. **(C)** VAE architecture. The encoder neural network learns estimates the posterior distribution of the latent variables, while the decoder learns a mapping from cell state to cell-wise transcriptional rates and simulates the generative process from an ODE system.

unspliced and spliced counts as continuous-valued functions of time, which is the first moment of a stochastic process [16] also known as the chemical master equation (CME). Thus, we investigated the discrete nature of RNA velocity by proposing a discrete model. First, we assume $u$ and $s$ are Poisson random variables at time $t=0$. Because the reactions are all uni-molecular, we apply the *t*heorem by Jahnke et al. [24] and reach a conclusion that both $u(t)$ and $s(t)$ are Poisson

random variables whose mean is given by the solution *to t*he continuous-valued ODE. Thus, we can fit the VeloVAE model with minimal change to the model architecture and simply use a Poisson likelihood to infer the latent variables and fit rate parameters.

In addition to the discrete model, we also extend previously developed theory to a binary mixture of ODE models at the gene level and also take advantage of the little-known fact that autoencoding variational Bayes allows inference of posterior distributions for parameters in the generative model (decoder network). This allows us to infer distributional estimates of the ODE rate parameters as well.

We train the model by maximizing the evidence lower bound of the marginal likelihood using mini-batch stochastic gradient descent [25]. Crucially, cells are loaded in batches during training, which means that not every cell must be used in each training iteration, dramatically decreasing both the time and memory requirements.

## 2.2 VeloVAE significantly improves model fit and latent time accuracy

We evaluated our method on a variety of real scRNA-seq datasets of different sizes and complexity [26–32] and compared our results with scVelo [10], UniTVelo [17], DeepVelo [18], VeloVI [20] and cellDancer [19]. UniTVelo did a comprehensive benchmarking tests on a number of other datasets [8,9,33–37]. Hence, we included them in our evaluation. In addition to continuous-count models, we also evaluated the discrete-count version of VeloVAE and PyroVelocity [21], since these are the only two methods directly modeling raw count numbers. We evaluated all models in terms of data reconstruction, how accurately they recover latent time and how well velocity flows match the biological truth. We also directly looked into each individual genes and compare the fitting qualitatively. Our results show that VeloVAE achieves comparable or better performance across all performance metrics, while existing methods are less versatile (Figs 2A, 2B, and S1).

To quantify model performance, we calculated mean squared error (MSE) for continuous-count models and log likelihood (LL) for discrete-count models (Figs 2B and 2C). For VeloVAE, VeloVI and PyroVelocity, we calculated these metrics on a held-out test set not used during training. Note that DeepVelo and UniTVelo cannot perform out-of-sample prediction, so we were not able to evaluate it on a held-out test set. ScVelo can in theory perform out of sample prediction, but this capability is not implemented in the scVelo package, so we calculated the metrics for scVelo on whole dataset in this comparison. CellDancer does not directly model count data and its training objective is cosine similarity, so we do not compute its MSE. Our results show that VeloVAE achieves comparable or better data fit than other methods in the evaluation. Furthermore, this better performance does not come from overfitting the training dataset - VeloVAE shows good performance on the held-out datasets.

To evaluate the accuracy of latent time inference, we used scRNA-seq datasets with cells sampled from multiple time points. These time points usually have rather coarse granularity (e.g., day 7 and day 14), and cells captured at the same time may span a wide range of developmental stages. Nevertheless, the inferred cell times should at least be correlated with the capture times. Thus, we computed the Spearman correlation between the cell times inferred by each method and the capture times (Fig 2A). VeloVAE consistently achieves comparable or better performance in terms of latent time accuracy, with scVelo often inferring latent time that is anticorrelated with real time (Fig 2A). Although scVelo and VeloVI infer latent time separately for each gene, they provide a post-hoc procedure for estimating a single global time for each cell. Using this global time for comparison with our methods casts these methods in the best possible light because the global time is more robust than the gene-specific latent times. The low time correlation from scVelo may be partly explained by inconsistency among the different notions of time fitted for each gene. To investigate this further, we computed the average time correlation between scVelo's and VeloVI's gene-specific and global latent time. As S2 Fig shows, the correlation between the global latent time and the latent time for each gene is indeed quite low; the latent time values for many genes are even anticorrelated with global latent time. We also note that cellDancer and DeepVelo do not include cell time in their models, but rather perform a post-hoc global time estimation based on velocity.

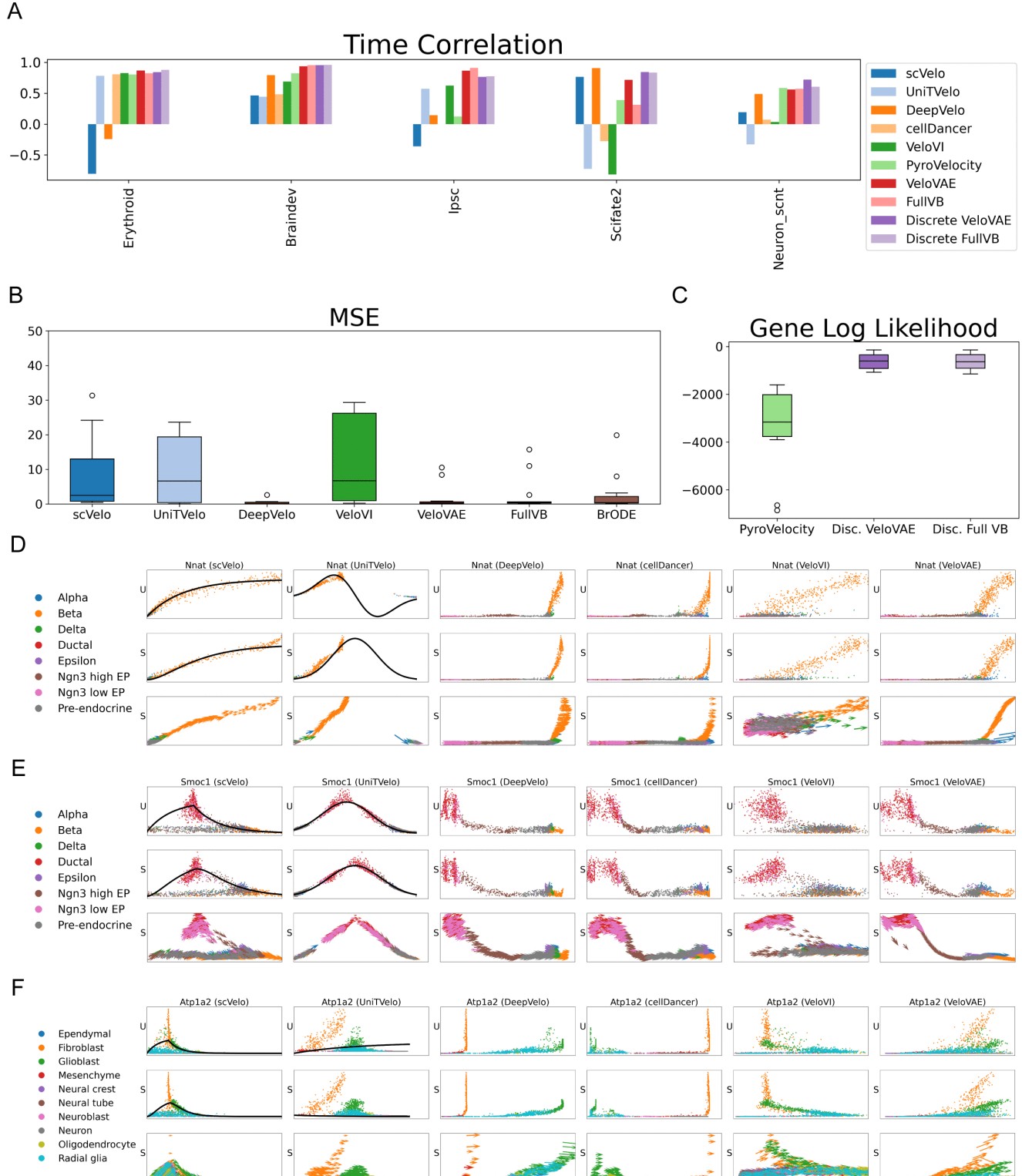

**Fig 2. VeloVAE significantly improves model fit and latent time accuracy and models complex gene expression dynamics (A)-(C) Quantitative performance comparison.** We report the correlation between inferred latent time and true capture time (when available) **(A)**, mean squared error (MSE) between the observed and predicted $(u,s)$ counts for continuous-count models **(B)** and gene log likelihood for discrete-count models **(C)** across

14 benchmarking datasets in a boxplot. **(D)-(F)** Examples of individual genes fit by different RNA velocity methods for a late-induced gene (D) and early-repressed gene (E) in pancreas and a gene with branching dynamics in subsampled mouse brain (F). Gene fits are shown for both $u$ and $s$ values, with inferred latent time on the x-axis. Fitted values from scVelo and UniTVelo are shown as lines, with observed data values shown as points. Only fitted values are shown for the other methods, because the fit is a point cloud (rather than a line) that would completely cover the observed data. Points arve colored by published cluster assignments, with the same colors as in the corresponding UMAP plots. The arrow length and direction indicate the velocity inferred for each cell.

To quantitatively measure how well the velocity flow matches our prior knowledge of different tissue developmental processes, we propose a modified version of cross-boundary direction correctness (CBDir). CBDir was initially introduced in the VeloAE paper [38]. This metric quantifies how accurately each estimated velocity matches prior knowledge at the cell type level. We generalized it by considering k-step neighbors and including time ordering in the evaluation. We also subtract a background CBDir under random walk to quantify how much better the model performs than random guessing. We compare our results in both gene expression space and 2D low-dimensional embedding based on umap (except for the human bone marrow dataset which based on tsne). Our results show that VeloVAE achieves a competitive performance on the benchmarking datasets (S1 Fig).

To further assess the reliability of VeloVAE inference, we investigated the time variance and cell state uncertainty across cells. We observe low time variance in early developmental cells, such as ductal cells in the pancreas dataset and progenitor cells in the human bone marrow dataset (S3 Fig). Correspondingly, these early-time cells exhibited higher cell state uncertainty compared with cells at later latent times (S4 Fig), consistent with biological expectations. That is, undifferentiated cells possess greater developmental potential and can diverge into multiple lineages. To better visualize the inferred dynamics, we extended the 2-D UMAP embedding by adding latent time as a third axis, generating a 3-D representation of the cellular state space along developmental progression. In this space, cells display smooth temporal transitions with clear lineage branching (S5 Fig), and 3-D velocity streamlines align with these trajectories (S6 Fig), confirming that VeloVAE captures coherent and biologically meaningful differentiation dynamics.

In addition, one of our benchmark datasets was generated with metabolic labeling to profile the embryonic differentiation process [31], providing a ground-truth reference for evaluating transcription rate parameters. We compare the transcription rates predicted by the VeloVAE continuous model with those from DeepVelo and cellDancer, the two other models that estimate cell-wise transcription rates. VeloVAE accurately infers transcription rates, achieving a mean Pearson correlation coefficient (PCC) of 0.72 between predicted and ground-truth transcription rates per cell per gene. Furthermore, VeloVAE exhibits a much higher correlation with the ground truth transcription rates compared to both DeepVelo and cellDancer (S7 Fig), demonstrating its superior performance in recovering cell-wise transcription rates. To enable a biologically meaningful comparison between predicted and ground-truth transcription rates, we further compute a Transcript-per-million-like (TPM-like) normalization (see Methods) and evaluate performance in terms of mean absolute error (MAE). Consistently, VeloVAE achieves the lowest MAE of approximately 13,046 transcripts per million, outperforming DeepVelo (44,450) and cellDancer (216,155).

## 2.3 VeloVAE better captures qualitative properties of complex gene expression dynamics

We designed the VeloVAE model to relax several of the restrictive assumptions of previous RNA velocity approaches. Thus, we expect that the model should show increased expressiveness, allowing it to capture qualitative properties of gene expression changes that previous approaches cannot. To assess this, we fit VeloVAE and all the other models on two representative datasets–mouse pancreas and brain–and inspected the resulting model fits. We found two types of qualitative behaviors that other existing approaches cannot accurately model, while VeloVAE can.

**2.3.1 Late induction and early repression.** We observed that genes with a late, short, or missing induction phase are particularly prone to being fit incorrectly by previous methods, due to each own limitation. First, for scVelo, the lack

of a common time scale, combined with the assumption that induction starts at $t=0$, also leads to frequent errors in estimating the overall direction of a gene (increasing or decreasing). Second, for UniTVelo, the absence of a mechanistic model results in unrealistic predictions. Last, velocity estimation methods like DeepVelo and cellDancer do not incorporate time into model fitting and thus, produce velocity that is inconsistent with human interpretation. A dataset from the mouse pancreas [26] illustrates these behaviors.

All methods in our study yield latent time values and stream plots for the pancreas dataset that are coherent with prior knowledge. However, by inspecting the fits for individual genes we can tell that previous methods either produce fitting that contradicts the biological truth or generate uninformative RNA velocity. Figs 2D and 2E show two sample genes, *Nnat* and *Smoc1*. The *Nnat* gene is not turned on until the pre-endocrine cells appear, whereas *Smoc1* is immediately switched off at the beginning of the differentiation process. We can see that scVelo, UniTVelo and VeloVI fail to detect late induction in *Nnat*. ScVelo assigns the latent time to zero for almost all cell types except for beta cells, while UniTVelo assigns some cells to the repression phase. The time inference from VeloVI is more accurate, but still mixes beta cells with endocrine progenitor cells. On the other hand, the post-hoc latent time from DeepVelo and cellDancer are consistent with the true developmental process, but the velocity does not match human interpretation that the mRNA count is increasing fast for *Nnat*. For *Smoc1*, scVelo and UniTVelo rearrange the order of progenitor and descendent cell types in an effort to force the gene to have a induction phase. The velocity from DeepVelo and cellDancer remains uninformative as there is no significant decreasing trend in the endocrine progenitor (Ngn3 low EP and high EP) cells. In contrast, VeloVAE not only correctly inferred the late induction pattern for *Nnat* and the early repression pattern for *Smoc1* but also produced perceptually correct velocity flow.

**2.3.2 Cell type bifurcations.** Many cell differentiation processes produce multiple descendant cell types from a single progenitor type, but previous RNA velocity approaches model only a single cell type. By including a cell-specific latent state, VeloVAE can model the continuous emergence of multiple cell types from a single progenitor type. For example, in the pancreas dataset, the *Nnat* gene is upregulated as cells differentiate toward the beta cell fate, but not in any other cell type (Fig 2D).

As another example, we fit all models on a developing mouse brain atlas [30]. For clarity, we subsampled the dataset to include only cell types arising from the neural tube and neural crest (see S8 Fig for an analysis of the full dataset).

In this system, neural tube cells develop into radial glia. Some of the radial glia cells differentiate into neuronal cell types, while the others give rise to the glial lineage, including glioblasts, oligodendrocytes, astrocytes, and ependymal cells. The neural crest cells develop to become fibroblast cells. In short, this is a complex system with many distinct lineages emerging. Nevertheless, VeloVAE can accurately model the complex, multi-lineage dynamics of genes in this system. For example, VeloVAE accurately models the behavior of the *Atp1a2* gene (Fig 2F), which is turned on in the fibroblast and glial cells with distinct transcription rates and remains off in the neuronal cells differentiated from radial glia. In contrast, scVelo rearranges the latent time values of the cells in a vain attempt to fit the *Atp1a2* gene into a single induction and repression cycle. VeloVI reverses the time order as fibroblast cells flow backwards to mesenchymal cells. We trained UniTVelo with unified cell time and the cell time is generally correct. However, without explicitly modeling branching, the parametric model from UniTVelo underfits the count data. Neither DeepVelo nor cellDancer accurately recovered the cell time for the fibroblast branch, and they suffer from the same problem of uninformative velocity.

Although VeloVAE accurately models bifurcations using a continuous cell state variable, the resulting parameters are not readily interpretable in terms of discrete cell types. Thus, we developed a model extension that aids in interpreting how the kinetic parameters of gene expression change across cell types. We extended the simple differential equation model shown in Fig 1 to a system of equations that we refer to as a branching ODE model (Fig 3A). Instead of a cell-specific transcription rate and fixed splicing and degradation rates, the branching ODE model assigns each cell type a unique ODE with cell-type-specific transcription, splicing, and degradation rates and an initial condition determined by the progenitor cell type (Methods). The model relies on a directed graph relationship among discrete cell types. The graph

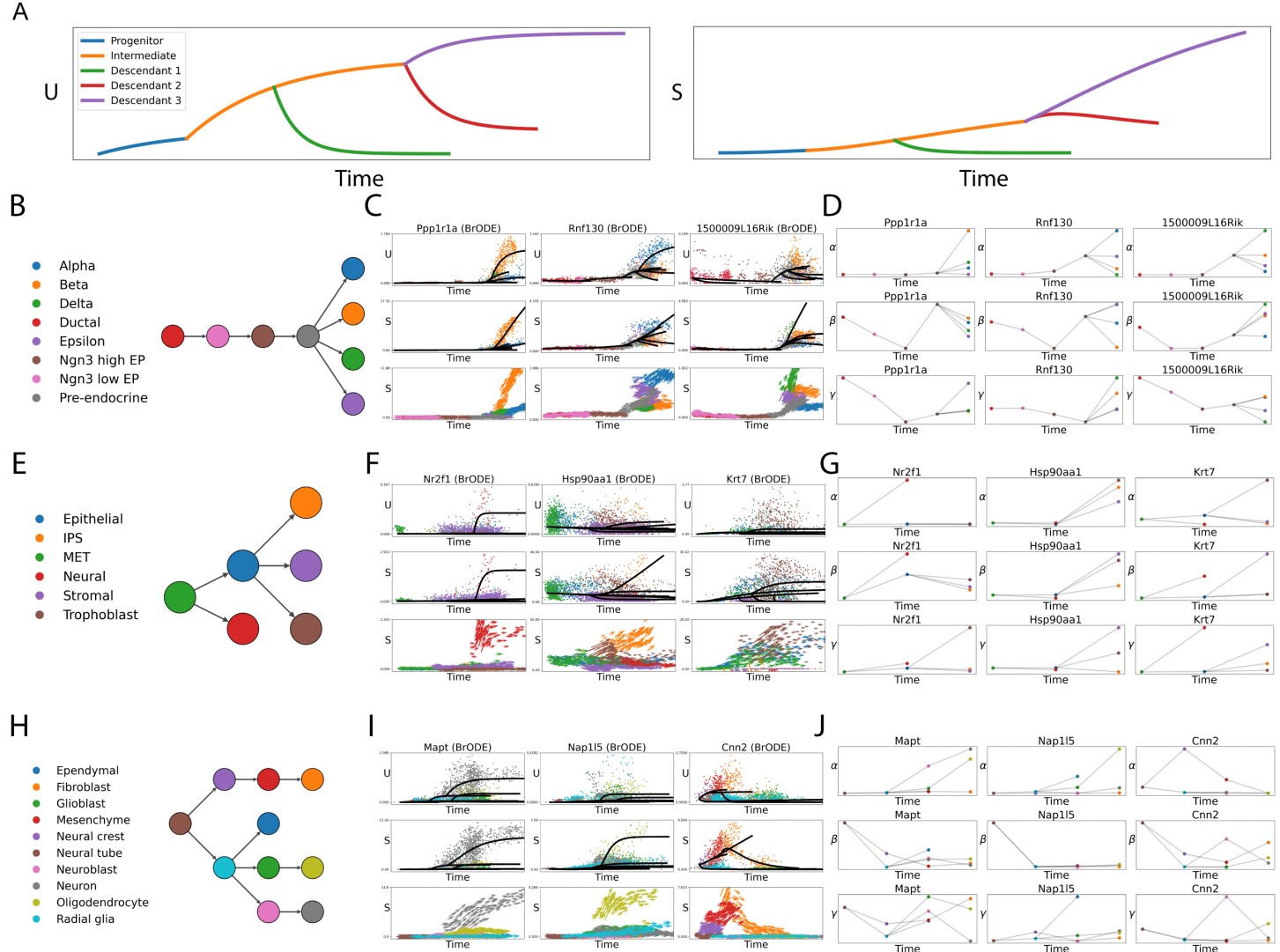

**Fig 3. Cell Type Transition Graph Inference and Branching Differential Equation Model. (A)** Schematic of branching ODE solutions. The example shows the output prediction of *u* and *s* versus time for a process with a single progenitor, an intermediate and three distinct descendant cell types. Inferred cell type transition graphs from pancreas **(B)**, iPSC **(E)** and subsampled mouse brain **(H)** datasets. In each graph, a vertex represents a cell type and a directed edge points from a progenitor cell type to its immediate descendant(s). Examples of individual genes fit by the branching ODE model for pancreas **(C)**, iPSC **(F)** and subsampled mouse brain **(I)** datasets. Each column represents a gene and plots the unspliced count, spliced count and RNA velocity versus time from top to bottom. The branching ODE fits are shown as solid lines. Cell-type-specific rate parameters inferred by branching ODE model for pancreas **(D)**, iPSC **(G)** and subsampled mouse brain **(J)** datasets are also shown. Each column represents a gene and plots the transcription ($\alpha$), splicing ($\beta$) and degradation ($\gamma$) rates from top to bottom. Each point represents a cell type, the types are arranged in chronological order, and progenitor-descendant relationships are indicated with lines.

can be inferred directly from the data, as we did here; if some aspects of the cell type transition graph are known, these can also be manually specified. We constrain the branching ODE model so that each cell type emerges at a specific time and the initial conditions of each cell type match the ODE prediction of its parent cell type at the time the child cell type emerges. This provides a qualitative view of the change of kinetic rates during cell development. We use the branching ODE model to replace the decoder of the VeloVAE, giving an alternative, more interpretable generative model.

Having established the accuracy of the results from VeloVAE, we used these results to infer cell type transition graphs. To infer the transition probability between cell types A and B, we used the cell times to simply count how often cells of type A and B occur in two immediately adjacent short time intervals (see Methods for details). We show three examples of these graphs (Figs 3B, 3E and 3H), inferred from the pancreas, iPS reprogramming, and mouse brain datasets from Fig 2. The inferred cell type transition graphs match closely with biological expectations for these systems, even for the complex mouse brain dataset.

To train the model, we first obtain the time and state assignments for each cell by training VeloVAE. Next, we infer a transition graph describing the progenitor-descendant relations among cell types (Methods). Finally, we fix the encoder of VeloVAE (so that latent time and cell state estimates are fixed) and estimate the parameters of the branching ODE model. We perform this parameter estimation by maximizing the Gaussian likelihood of all genes under the branching ODE model, which is equivalent to minimizing the Mahalanobis distance between model fit and observed data.

We trained branching ODEs on pancreas, iPSC and subsampled mouse brain datasets. The cell type transition graphs are all consistent with prior knowledge (Figs 3B, 3E and 3H). For example, our computational method successfully finds the expected cell differentiation path in pancreatic development, starting from ductal cells and branching into $\alpha$, $\beta$, $\delta$ and $\epsilon$ cells (Fig 3B).

In addition, the branching ODE is able to infer different and asynchronous gene expression kinetics in multiple branches and successfully captures different rates in different branches in these datasets. For example, *Ppp1r1a* has almost all transcription activity in $\beta$ cells (Fig 3C), which was verified by previous studies [39,40]. The *Rnf130* and *1500009L16Rik* genes similarly show significant branching trends that are accurately modeled by the branching ODE (Fig 3C). The branching ODE model also accurately fits genes that show differential kinetics among lineages that emerge during induced pluripotent stem cell reprogramming. For example, *Nr2f1* is strongly and specifically upregulated in neural-like cells (Fig 3F). *Hsp90aa1* is upregulated moderately in Trophoblast and strongly in IPS cells. *Krt7* is upregulated in epithelial-like and trophoblast-like cells with differing expression levels (Fig 3F).

Another example is *Mapt* from the mouse brain dataset (Fig 3I). The gene is upregulated strongly in neurons and subsequently transcribed at much lower levels in oligodendrocytes, which coheres with previous studies [41]. The *Napl15* gene shows the opposite trend, with high transcription in oligodendrocytes and low but detectable transcription in glioblasts and neurons (Fig 3I). As another example, the *Cnn2* gene is most highly transcribed in the early transition from neural crest to mesenchyme and then to fibroblast (Fig 3I). The cell-type-specific rate parameters inferred by fitting the branching ODE describe the differences in transcription, splicing, and degradation that cells undergo as they differentiate (Figs 3D, 3G and 3J).

## 2.4 VeloVAE accurately models human hematopoiesis from discrete count data

Previous papers have noted that hematopoiesis is a particularly difficult system for existing RNA velocity methods [16]. Latent time and velocity inferences often seem to point in the opposite direction of the known blood cell differentiation hierarchy. Two aspects in particular likely make this system challenging. First, many distinct cell types emerge simultaneously from the hematopoietic stem cell (HSC). Recent studies suggest that hematopoietic progenitors are more like a continuum of primed states than a set of discrete states neatly organized in a tree structure [42]. Second, blood cells are produced exceptionally rapidly compared with other cell types; a recent study estimated that about 2 million new red blood cells per second enter the bloodstream [43]. Thus, blood cells may use special gene regulatory mechanisms such as "transcriptional boosts" and other time-varying rate parameters.

To investigate whether VeloVAE can resolve these difficulties, we analyzed a recent human bone marrow dataset [29]. Our stream plot shows that VeloVAE correctly identifies HSCs as the start of differentiation and predicts that they differentiate into megakaryocytes, platelets, dendritic cells, monocytes, and B-cells (Fig 4A). As a quantitative comparison, we computed k-step CBDir (See Methods for the definition) on RNA velocity in the original gene expression space. Our

**Fig 4. VeloVAE Models Complex Kinetics in Bone Marrow Cells. (A)** UMAP plots colored by published cell types overlaid with velocity inferred by all methods. **(B)** Quantitative performance comparison. We computed k-step cross-boundary direction correctness (K-CBDir) for k = 1,2,3,4,5. **(C)** Examples of individual genes fit by all methods. Gene fits are shown for both *u* and *s* values, with inferred latent time on the x-axis. Similar to Fig 2, fitted values

from scVelo and UniTVelo are shown as lines, with observed data values shown as points and only fitted values are plotted for other methods. Points are colored by published cluster assignments, with the same colors as in the corresponding UMAP plots. The arrow length and direction indicate the velocity inferred for each cell. **(D)** Examples of low-count genes fit by discrete VeloVAE. **(E)** UMAP plots colored by latent time inferred by different methods.

results show that all variants of VeloVAE perform equally well or better than other methods in terms of velocity direction (Fig 4B). Note that both velocity stream plot and CBDir are high-level representations of cellular dynamics. In fact, the quality of gene-level velocity fitting is not embodied entirely in the stream plot. Inspecting the fits of individual genes revealed that other methods fail to capture the complexity of multiple lineages simultaneously emerging. For example, the *CD36* gene is upregulated strongly in red blood cells, moderately in monocytes, and slightly in dendritic cells. VeloVAE correctly models the expression dynamics while other methods either reverse the cell order or fail to capture distinct dynamics (Fig 4C). Noticeably, the discrete VeloVAE model achieves the best performance and we show that it is capable of fitting genes with very low count numbers (Fig 4D). VeloVAE accurately infers the true differentiation trajectory in this complex system, but none of the other methods do (Fig 4E). In the latent time inference, scVelo, UniTVelo, and DeepVelo identify CD14 monocytes as having the earliest latent time, whereas CD14 monocytes are known to represent terminally differentiated cells in hematopoiesis. Similarly, cellDancer infers memory B cells as the earliest cell population, and VeloVI identifies plasmablasts as root cells. In contrast, VeloVAE correctly assigns HSCs as the population with the earliest latent time. Additionally, VeloVAE identifies naive CD4 and CD8 T cells as distinct developmental roots, which subsequently differentiate into CD8 memory and CD8 effector T cells. Accurately inferring these developmental paths is critical for evaluating method performance. As shown in Figs 4A and 4E, VeloVAE correctly reconstructs the ground truth trajectories from HSCs→LMPP→GMP→CD14/CD16 monocytes, as well as the HSC→RBC lineage. In contrast, scVelo, UniTVelo, DeepVelo, and cellDancer predict the RBC lineage in reverse, from RBCs toward HSCs, while VeloVI predicts the CD14/CD16 lineage in reverse. Overall, VeloVAE is the only model that accurately infers differentiation paths consistent with prior biological knowledge.

## 2.5 VeloVAE accurately models organ development in whole mouse embryos

The challenges of studying cell differentiation with single-cell data become particularly acute at the scale of an entire organism. For example, Cao et al. performed single-cell RNA-seq on 61 entire mouse embryos sampled from E9.5-E13.5, the developmental period when mouse organogenesis occurs [4]. The dataset contains 1,191,071 cells after preprocessing, which fall into 38 major cell types and can divided into 9 major lineages. Pseudotime analysis and RNA velocity analysis have been performed on this dataset, but both analyses were highly manual processes that required separate curation of dozens of cell subsets [4,44].

In principle, the continuous cell state variable of VeloVAE is sufficiently expressive to provide a single model of the differentiation potential for an entire organism. To investigate this, we trained VeloVAE on the entire mouse organogenesis dataset. Because of the large size and cellular diversity of this dataset, we used a larger batch size of 1024 and increased the dimension of **c** from 5 to 30. Our results show that VeloVAE discovered a meaningful latent cell state space representing the whole developmental process (Fig 5A). To examine the results in more detail, we ran UMAP individually on the same 9 broad cell lineages that were identified in the initial paper. Importantly, we performed this subset analysis purely for visualization purposes–the cell time, cell state, and kinetic rate parameters were estimated only once using all cells jointly (S8-S14 Figs).

These visualizations indicate that latent time and velocity estimates are highly consistent with cell capture times and biological prior knowledge (Figs 5B-E, S9 and S10). Remarkably, the latent time estimates are even more accurate than the authors' reported pseudotime values in several cases. For example, the VeloVAE latent time estimates for the two lineages with the most cells, mesenchyme and neural tube, both show a higher correlation with capture time than the pseudotime estimates reported by

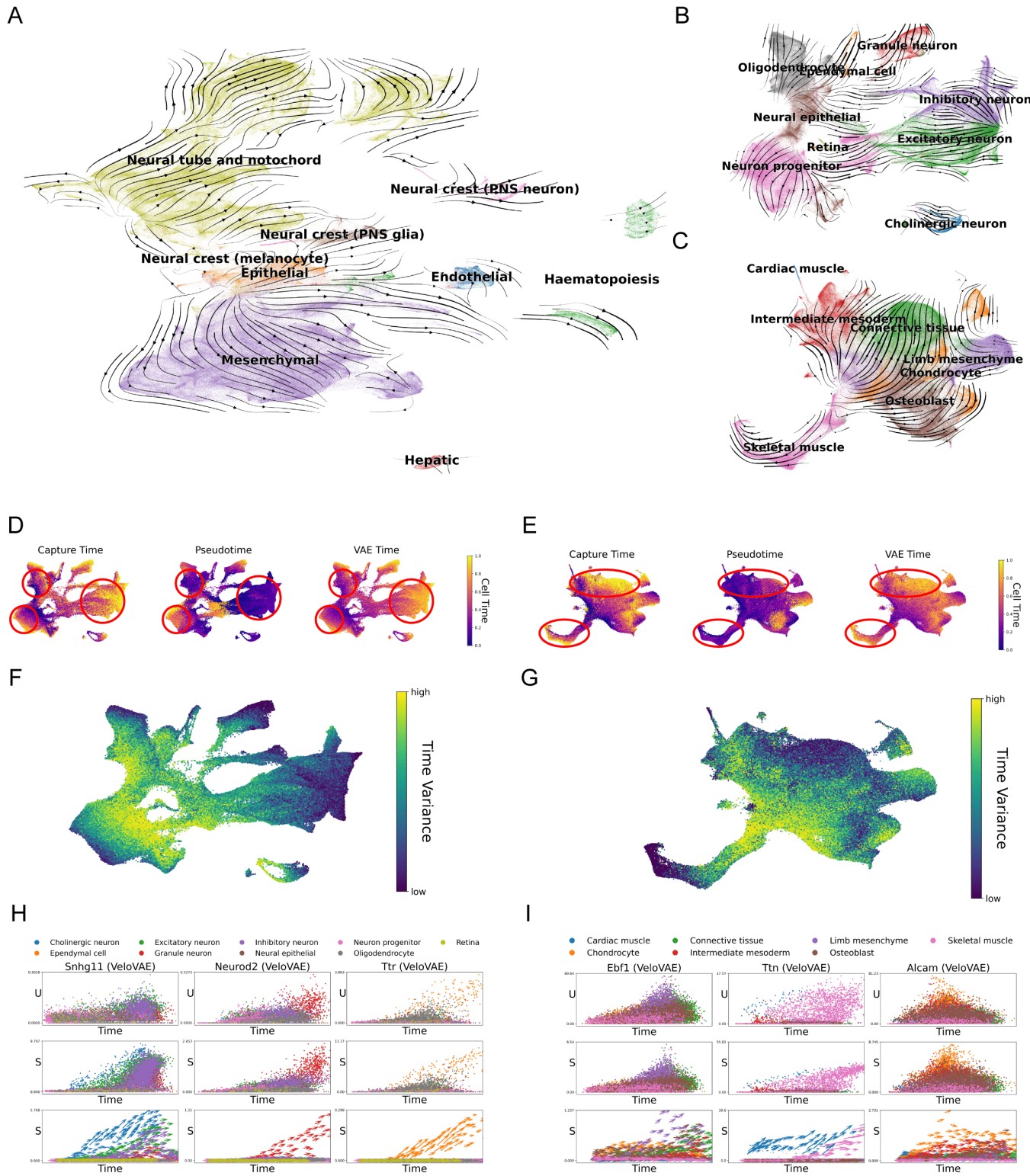

**Fig 5. Whole mouse embryo analysis. (A)** UMAP of entire mouse embryo dataset, colored by broad cell class. **(B)-(C)** UMAP plots colored by published cell types overlaid with velocity inferred by VeloVAE for neuronal **(B)** and mesenchymal **(C)** trajectories. The UMAP coordinates are computed using only cells from each trajectory, but the velocity estimates are from a single VeloVAE model fit on all cells. **(D)-(E)** UMAP plots colored by capture

time, published pseudotime, and VeloVAE latent time values for the neuronal **(D)** and mensenchymal **(E)** trajectories. Note that latent times are from a single VeloVAE model fit on all cells. **(F)-(G)** t-SNE plots colored by **(F)** cell state uncertainty and **(G)** time variance. **(H)-(I)** VeloVAE fits for selected genes from the neuronal **(H)** and mesenchymal **(I)** trajectories.

Cao et al [4]. In particular, the pseudotime estimates for the oligodendrocyte, neural progenitor and inhibitory and excitatory neuron cells are essentially uncorrelated with capture time, whereas the VeloVAE results proceed in the direction of increasing embryonic stage (Fig 5D). Similarly, the pseudotime estimates for skeletal muscle cells are actually uncorrelated with cell capture times, and the pseudotimes of developing connective tissue cells substantially underestimate their developmental progress (Fig 5E).

These results are remarkable because the pseudotime values were generated by the developers of Monocle3–one of the most popular pseudotime methods–on their own data using their own tool in a highly manual process. Cao et al. used expert knowledge to group cells into "subtrajectories" and choose root cells for each. In contrast, we obtained the VeloVAE results by simply running the model jointly on all cells with no manual curation. The accuracy of the VeloVAE latent time estimates underscores the power of this approach for studying cell differentiation in large, complex, multi-lineage single-cell datasets. This comparison also highlights the inherent difficulty of pseudotime analysis in such complex datasets, particularly when the continuous nature of differentiation and the presence of multiple "root cell" populations make it very challenging to divide the cells into discrete subtrajectories.

In comparing latent time and pseudotime values, we realized that the latent time values offer another advantage: they can be assigned real time units. Because we use the capture times as prior information, this places the inferred latent time on the same scale as the capture times. That is, with appropriate normalization, we can report the latent times in units of days or hours. Consequently, the rate parameters can also be assigned units, such as transcripts per minute. To explore this, we converted the latent time values into units of minutes and normalized the transcription rate parameters to the absolute number of transcripts in a typical mammalian cell. This shows that the

transcription rates are roughly on the order of 1–10 transcripts per minute for most genes (S15 Fig), in accordance with previous estimates from single-molecule FISH experiments [45]. The splicing rates are predicted to be slightly lower, on the order of 0.1-1 transcripts per minute. This is consistent with a previous study that estimated the elongation rate in human cells at 3.8 kb/min and found that splicing begins within 10 min [46]. Our estimated degradation rates are similar to the transcription rates, on the order of 1–10 transcripts per minute. Though these estimates should not be taken as definitive, we find it reassuring that our estimated rate parameters are on roughly the correct order of magnitude compared with prior knowledge.

## 3  Discussion

The VeloVAE model uses variational Bayesian inference to estimate cell differentiation progress, cell state, and kinetic rate parameters in a statistically principled fashion. Our approach not only improves gene fitting in many cases, but also resolves some of the key limitations of previous RNA velocity methods. The expressiveness of neural networks makes it possible to adapt the model to various types of gene expression kinetics in biological differentiation systems of varying complexity. Moreover, this model expressiveness does not come at the cost of interpretability; we still learn a set of rate parameters with direct biophysical interpretations. Another key advantage of VeloVAE is that it can perform inference using only the unspliced and spliced mRNA counts from a set of cells, but is also able to incorporate additional prior information such as cell capture times when available. Additionally, the use of mini-batch stochastic gradient descent allows the method to process large datasets without loading all cells into memory at once.

These advantages of VeloVAE make the RNA velocity results much more useful and interpretable in several ways. First, the latent times inferred by the model are sufficiently accurate that they can be used to order cells according to differentiation progress–with even higher accuracy than pseudotime inference in some cases. Second, the ability to use cell capture times as a prior distribution places the inferred cell times on a time scale with known units (e.g., hours or days). Knowing the time

units also gives the kinetic rate parameters a more direct interpretation. As more time-series scRNA-seq datasets become available, our time-prior framework will be increasingly valuable for inferring diverse dynamic biological processes. Moreover, it can facilitate the transfer of temporal information to non–time-series datasets, providing meaningful latent time estimates even in the absence of explicit experimental timing. Furthermore, the fitted and extrapolated values are now qualitatively accurate for individual genes–in contrast to results from previous RNA velocity methods, where the individual gene fits are often poor even when the global latent time and stream plots are reasonable. This opens up many new applications in which knowing the direction and rate of change for an individual gene is important, rather than simply a qualitative, global visualization. For example, one could now perform direct comparison of the times at which different genes are transcribed, revealing the relative order in which genes are activated. Additionally, we showed that the latent time results and future cell state predictions are sufficiently accurate that they can be used to infer transition graphs among cell types in many cases. Finally, our branching ODE model provides important insights into how transcription, splicing, and degradation rates change during cell type bifurcations, which holds promise for understanding the factors regulating cell fate decisions.

Despite the advantages of VeloVAE for velocity inference, the current model still has limitations. As a Bayesian model, VeloVAE is sensitive to choice of prior and hyperparameters. Incorporating more biologically informed priors (like basal transcription rate for different genes) could potentially improve the performance of the velocity inference in the future. As with other RNA velocity methods, VeloVAE is also affected by the quality of spliced and unspliced count quantification. Developing more accurate preprocessing and quantification tools will help produce more reliable and biologically meaningful velocity inference results. Overall, VeloVAE provides a powerful and flexible probabilistic framework for modeling transcriptional dynamics at single-cell resolution. By integrating Bayesian inference with dynamical modeling, it enables both accurate velocity estimation and uncertainty quantification, offering deeper insights into gene regulation and cell-state transitions. Future extensions incorporating biologically informed priors and improved preprocessing will further enhance its robustness and applicability across diverse biological systems.

## 4 Code and data availability

Our code is available at https://github.com/welch-lab/VeloVAE. We also provide the reproducible notebooks and scripts for all the benchmark methods and datasets. All source code, including the implementations and hyperparameter configurations for each benchmark method, is also available at the github repository. For the datasets used in this paper, Pancreas, Dentate Gyrus, Mouse Erythroid and Bone marrow datasets could be obtained from scVelo api using scv.datasets. The count matrix for the Mouse Hindbrain (pons) dataset is obtained from Kharchenko Lab. The sequencing data for the mouse brain and Mouse Hindbrain (pons) datasets could be accessed through PRJNA637987 and PRJNA637987. The sequencing data for the other datasets used in this study could be accessed through GEO: IPSC (GSE122662), Human Hematopoiesis (GSE102698), Human Erythroid (GSE167576), Mouse Retina (GSM3466902), Intestinal Organoid (GSE128365), Neurogenesis (GSE141851), Scifate2 (GSE236520) and Tracer (GSE220949).

## 5 Methods

### 5.1 Problem setup

First, we formulate the computational problem of modeling cellular gene expression changes using the mathematical language. Single-cell RNA experiments produce cell-by-gene count matrices. RNA reads are further classified to be either spliced or unspliced, and correspondingly, there are two count matrices of size $n \times m$, representing unspliced and spliced RNA counts. Each sample (cell), indexed by $i$, is represented by a vector $X_i(t_i) \in \mathbb{R}^m$ parametrized by unobserved time $t_i$. The trajectory $X_i(t)$ is governed by some differential equation plus random noise. Our goal is to infer the cell time $t_i$ and predict the velocity $\frac{dX_i}{dt}$.

We list a set of definitions from our previous work [23] for convenience of description. We refer the reader to the paper for more details.

**Definition 1. (Gu et al., 2022 [23]).** *Let $u_g$ and $s_g$ denote the unspliced and spliced mRNA count of the g-th gene. Let $\mathcal{G} = \{1, 2, \ldots, G\}$ be a set of genes measured in an scRNA-seq experiment. The* **feature vector** *of a cell is defined as* $\mathbf{x} = [u_1, u_2, \ldots, u_G, s_1, s_2 \ldots, s_G]^T$.

**Definition 2. (Gu et al., 2022 [23]).** *The* **kinetic equation** *of gene g is defined as a system of ordinary differential equations relating changes in u and s over time. If there exists a solution $F(t; \theta)$ to the initial value problem with $u(0) = u_0, s(0) = s_0$, we call this solution the* **kinetic function** *for g.*

**Definition 3. (La Manno et al., 2018 [9]).** *Given a kinetic function u(t) and s(t) of a gene, the* **RNA velocity** *of the gene is defined as* $\frac{ds}{dt}$.

## 5.2 Modeling gene expression kinetics

As first proposed in 2018 [9], the kinetic equation is modeled by a system of two linear ODEs:

$$\frac{du}{dt} = \alpha I_{\{t < t_{off}\}} - \beta u, \quad \frac{ds}{dt} = \beta u - \gamma s, \tag{1}$$

where $I_{\{\cdot\}}$ is an indicator function for the condition in brackets. The model parameters $\alpha$, $\beta$ and $\gamma$ correspond to the RNA transcription, splicing and degradation rates, respectively. The model assumes that two discrete phases can occur in the gene expression process: (1) induction, when new unspliced RNA molecules are being transcribed (2) repression, when the transcription process stops and no new unspliced molecules are made. The induction phase is assumed to start at $t_{on}=0$ and the transition from induction to repression occurs at time $t_{off}$. Given an initial condition $u(0) = u_0, s(0) = s_0$, the analytical solution to the ODE is

$$u(t) = u_0 \exp(-\beta\tau) + \frac{\widetilde{\alpha}}{\beta}\left(1 - \exp(-\beta\tau)\right) \tag{2}$$

$$s(t) = s_0 \exp(-\gamma\tau) + \frac{\widetilde{\alpha}}{\gamma}\left(1 - \exp(-\gamma\tau)\right) + \frac{\widetilde{\alpha} - \beta u_0}{\gamma - \beta}\left(\exp(-\gamma\tau) - \exp(-\beta\tau)\right)$$

$$\widetilde{\alpha} := \alpha I_{\{t < t_{off}\}}, \quad \tau := t I_{\{t < t_{off}\}} + (t - t_{off}) I_{\{t \geq t_{off}\}} \tag{3}$$

## 5.3 BasisVAE

BasisVAE [22] was orignally designed to account for feature-level clustering. It aims at clustering different features (dimensions) of a high-dimensional observation $x = [x_1, x_2, \ldots, x_d]^T \in \mathbb{R}^d$ via a variational auto-encoder. A potential application is to cluster genes with similar patterns.

The model assumes each feature is generated by one of $K$ basis functions, i.e.,

$$x_i = \sum_{k=1}^{K} w_k^{(i)} f_k(\mathbf{z}_i), \quad f_k : \mathbb{R}^l \to \mathbb{R}^d, i \in \{1, 2, \ldots, d\} \tag{4}$$

Here, $\mathbf{z}$ is a low-dimensional latent variable generating the observation $x_i$. The clustering information is stored in $\mathbf{w}_i = [w_1^{(i)}, \ldots, w_K^{(i)}]^T \sim \text{Categorical}(\pi_1, \ldots, \pi_K)$ and $\boldsymbol{\pi} := [\pi_1, \ldots, \pi_K] \sim \text{Dirichlet}(\psi_1, \ldots, \psi_K)$. Thus, there are two additional latent variables, $\mathbf{w}$ and $\boldsymbol{\pi}$. According to Märtens and Yau [22], we can marginalize $\boldsymbol{\pi}$ using collapsed inference. The collapsed evidence lower bound (ELBO) is given by

$$\mathcal{L} = \sum_{j=1}^{n} \mathbb{E}_{q(\mathbf{z}_j|\mathbf{x}_j)} \mathbb{E}_{q(\mathbf{w})} \log p\left(\mathbf{x}_j|\mathbf{z}_j, \mathbf{w}\right)$$

$$+ \log \left( \int \exp\left(\mathbb{E}_{q(\mathbf{w})}\left[p(\mathbf{w}|\pi)\right]\right) p(\pi) \right)$$

$$- \mathbb{E}_{q(\mathbf{w})}[\log q(\mathbf{w})]$$

$$- \sum_{j=1}^{n} KL(q(\mathbf{z}_j|\mathbf{x}_j)||p(\mathbf{z}_j))$$

(5)

Here, the second term has an analytical form of

$$\log\left[\frac{\Gamma(\sum_k \psi_k)\prod_k \Gamma(n_k + \psi_k)}{\prod_k \Gamma(\psi_k)\Gamma(\sum_k(n_k + \psi_k))}\right]$$

(6)

where $n_k := \sum_{i=1}^{d} \psi_k^{(i)}$ and $\Gamma(\cdot)$ is the gamma function.

### 5.4 Variational mixture of ODE model

**5.4.1 ODE formulation.** We extend the ODE formulation (1) to broader cases. First, we adopt the idea of varying transcription rate $\rho$ proposed in our previous work [23]:

$$\frac{du}{dt} = \rho\alpha - \beta u, \quad \frac{ds}{dt} = \beta u - \gamma s, \quad u(0) = s(0) = 0$$

(7)

To account for repression-only genes, we add a second set of ODEs:

$$\frac{du}{dt} = -\beta u, \quad \frac{ds}{dt} = \beta u - \gamma s, \quad u(0) = u_0 > 0, \quad s(0) = s_0 > 0$$

(8)

Let $(u_g^{ind}(t), s_g^{ind}(t))$ and $(u_g^{rep}(t), s_g^{rep}(t))$ be the analytical solutions to equations (7) and (8) of the $g$-th gene. Define $\boldsymbol{F}_{ind} = [u_1^{ind}(t), \ldots, u_G^{ind}(t), s_1^{ind}(t), \ldots, s_G^{ind}(t)]^T$ and $\boldsymbol{F}_{rep} = [u_1^{rep}(t), \ldots, u_G^{rep}(t), s_1^{rep}(t), \ldots, s_G^{rep}(t)]^T$. Given $t$, $\boldsymbol{F}_{ind}$ and $\boldsymbol{F}_{rep}$ are vectors containing solutions to two sets of kinetics equations with different assumptions. As we will discuss below, solutions to the two sets of ODEs are two basis functions of our VAE model.

**5.4.2 Generative process.** The generative process for the variational mixture of ODE model is as follows:

$$t \sim \mathcal{N}(t_0, \sigma_0^2), \quad \mathbf{z} \sim \mathcal{N}(\mathbf{0}, \mathbf{I})$$

$$\pi \sim \text{Dirichlet}(\psi)$$

$$\mathbf{w} \sim \text{Categorical}(\pi)$$

$$\tilde{\alpha} = \rho \odot \alpha, \quad \rho = g(\mathbf{z}; \theta_\rho)$$

$$\mathbf{x_{ind}} \sim \mathcal{N}(F_{ind}(t; \theta), \Sigma_\mathbf{r})$$

$$\mathbf{x_{rep}} \sim \mathcal{N}(F_{rep}(t; \theta), \Sigma_\mathbf{r})$$

$$\mathbf{x} = \boldsymbol{I}_{\{\mathbf{w}=1\}} \odot \mathbf{x_{ind}} + \boldsymbol{I}_{\{\mathbf{w}=2\}} \odot \mathbf{x_{rep}}$$

(9)

Here, $g(\cdot)$ is a neural network with parameters $\theta_\rho$, $\boldsymbol{I}_{\{\cdot\}}$ is a vector indicator function, $\odot$ is the elementwise product, $F$ is the kinetic function of all genes, and $\Sigma_\mathbf{r}$ is a diagonal covariance matrix.

In the context of BasisVAE, $F_{ind}$ and $F_{rep}$ are two basis functions representing two potential data generative processes. By introducing $\boldsymbol{w}$, we distinguish genes by different generative processes, thus promoting gene-level clustering.

**5.4.3 Parameter inference and neural network architecture.** The inference process uses variational approximation. For the cell time $t$ and cell state $\boldsymbol{z}$, we assume a Gaussian posterior, as in [23]. For $\boldsymbol{w}$, we initialize its variational distribution $q(\boldsymbol{w}; \pi)$ by using the steady-state [9] and dynamical models [10]. In addition, we performed gene-level clustering to robustly estimate $\pi$. See the section entitled "Parameter Initialization" for details. The posterior parameters $\pi$ is updated during training.

We train VeloVAE using the standard mini-batch stochastic gradient descent. The training objective function is the collapsed evidence lower bound (ELBO) (5). We assume the prior $p(\boldsymbol{z},t)=p(\boldsymbol{z})p(t)$ is a multi-variate Gaussian distribution where $p(\boldsymbol{z})$ is isotropic Gaussian and $p(t) \sim \mathcal{N}(t_0, \sigma_0^2)$.

The encoder of VeloVAE is a multi-layer perceptron (MLP) with 2 hidden layers and 4 output layers representing the variational posterior mean and standard deviation of $\boldsymbol{z}$ and $t$. We use an MLP that is the mirror image of $h$ (two layers with 250 and 500 neurons, respectively) to learn the mapping $g$ from $\boldsymbol{z}$ to $\rho$. We use mini-batch stochastic gradient descent (SGD) to estimate the ODE rate parameters. There is a second stage of training where we estimate and iteratively update an initial condition $u(t_0) = u_0, s(t_0) = s_0$ for each cell to obtain a more accurate velocity inference. We list all critical hyper-parameters in S1 Table, the default hyper-parameters setting for Branching ODE is listed in S2 Table. VeloVAE employs an early-stopping criterion based on validation likelihood to prevent overfitting and to automatically determine the optimal training epoch. When time priors (e.g., capture times) are available, they are incorporated during training, which improves both latent time estimation and velocity inference. For hyperparameter selection, we adopted a consistent search strategy across datasets. The main hyperparameters tuned include the learning rate and the dimensionality of the latent space. Based on empirical observations, the learning rate for the ODE solver was set to be ten times larger than that of the neural network optimizer (typically 1e-3 for the ODE and 1e-4 for the neural network), which enhanced convergence speed and numerical stability. The latent space dimension was adjusted according to the complexity of each dataset ranging from 5 for simpler trajectories to 30 for more heterogeneous datasets. The final hyperparameters used for each benckmark dataset are summarized in S3 Table.

**5.4.4 Initial conditions.** Because each cell potentially has different ODE parameters, determining the initial conditions is more complex than in the scVelo model. Thus, we train the model with a simple assumption that $u_0 = s_0 = 0$ for the inductive ODE model and $u_0 > 0, s_0 > 0$ for the repressive ODE model in all of our experiments. This still yields excellent data reconstruction and latent time inference. However, the initial conditions are important for accurately predicting the future state of each cell. To improve the accuracy of future state prediction, we first train the VeloVAE to convergence using default initial conditions of two ODE settings so that latent times and cell states are accurate, then determine the initial conditions for a cell at time $t$ by simply averaging the $(u,s)$ values observed in an immediately preceding time interval $[t - \delta_1, t - \delta_2]$. We then fine-tune the ODE parameters using these updated initial conditions, keeping latent time and cell state fixed.

**5.4.5 Parameter initialization.** The neural network weights are randomly initialized with Xavier uniform distributions [47]. The only exception is the output layer of the decoder network, which is initialized with Xavier normal distributions, to prevent gradient vanishing of the sigmoid activation. VeloVAE applies the same initialization method as scVelo by applying the steady-state assumption and the dynamical model. Next, a global cell time is estimated as the median of locally initialized time across all genes.

If capture times are available, VeloVAE initializes a unique cell time by directly sampling from a distribution centered at the real time. Whether the initial global time is estimated by the steady-state model or directly sampling from capture time, rate parameters are always re-estimated based on the dynamical model. We still make the assumption that $\beta = 1$. First, switch-off time is estimated as the average global cell time of the steady-state cells. Next, we estimate $\alpha$ and the switch-on time $t_{on}$ by solving a set of equations using two cells with their initialized cell time $(u_1, t_1), (u_2, t_2)$:

$$u_1 = \frac{\alpha}{\beta}\left(1 - e^{-\beta(t_1-t_{on})}\right)$$

$$u_2 = \frac{\alpha}{\beta}\left(1 - e^{-\beta(t_2-t_{on})}\right)$$

$$\implies t_{on} = \frac{\log(u_1-u_2)}{u_1 e^{-t_2} - u_2 e^{-t_1}}, \quad \alpha = \frac{\beta u_1}{1 - e^{-\beta(t_1-t_{on})}}$$

$$(10)$$

Note that $u_1$ and $u_2$ are chosen as the sample average around the median and top quantile of $u$ values to promote robustness against noise. Finally, $\gamma$ is estimated by $\frac{\alpha}{s_{ss}}$ where $s_{ss}$ is the estimated steady-state $s$ value.

Let $n_{ind}$ and $n$ be the number of cells in the induction phase and total number of cells. A naive approach will be $\pi_1 = \frac{n_{ind}}{n}$, i.e., assigning the proportional of cells in the induction phase. However, such a gene-wise initialization is sensitive to data noise, similar to the problem of gene-wise time fitting. Instead, we make use of a simple but critical fact: genes generated by the same underlying generative process (inductive or repressive) should have similar dynamical behavior over time. This means the count number should be positively correlated when both genes are generated by the same basis function. Using this insight, we compute a gene-by-gene weighted adjacency matrix $\mathbf{A}$ where $A_{ij} = \frac{1}{2}[corr(\mathbf{U}_{:,i}, \mathbf{U}_{:,j}) + corr(\mathbf{S}_{:,i}, \mathbf{S}_{:,j})] + 1$, where $corr(\cdot, \cdot)$ denotes Pearson correlation coefficient and $\mathbf{U}_{:,i}, \mathbf{S}_{:,i}$ denotes unspliced and spliced counts of the $i$-th gene across all cells. We apply spectral clustering on genes using $\mathbf{A}$. The number of clusters is determined by using an empirical threshold on the singular values of $\mathbf{A}$ with a Gaussian noise assumption [48]. Denote $\tilde{w}_g$ as the proportion of cells in the induction phase and $y_g \in \{1, 2, \ldots, K_y\}$ be the clustering result of the $g$-th gene. Next, for each cluster $y$, we perform two Kolmogorov-Smirnov tests between $\mathcal{W}_y = \{w_g : y_g = y\}$ and Dirichlet(5.0,5.0). The two null hypotheses are

1. $\mathcal{W}_y$ is sampled from Dirichlet($\psi_1, \psi_2$) whose CDF is less than CDF of Dirichlet(5.0,5.0)

2. $\mathcal{W}_y$ is sampled from Dirichlet($\psi_1, \psi_2$) whose CDF is greater than CDF of Dirichlet(5.0,5.0)

If the first null hypothesis is not rejected, genes in cluster $y$ are likely to be inductive. We set $\psi_1 > \psi_2$ and reinitialize $\mathcal{W}_y$ by sampling from Dirichlet($\psi_1, \psi_2$). If the second null hypothesis is not rejected, genes in the cluster $y$ are likely to be repressive and we perform the same sampling but with $\psi_1 < \psi_2$. If both null hypotheses are rejected, we sample $\mathcal{W}_y$ from a mixture of two Dirichlet distributions mentioned in the previous two cases. To prevent a bad initialization, which is likely when the dynamical model fails, we add an option to replace $\boldsymbol{\pi} = [\pi_1, \pi_2]^T$ by $\boldsymbol{\pi} = [\pi_2, \pi_1]^T$. This helped to correct the time inference in some of our experiments.

**5.4.6 Estimating ODE parameter uncertainty.** The VAE model can be extended [49] to account for uncertainty in the parameters of the generative model, which in our case are the kinetic rate parameters of the ODE system. Let $\boldsymbol{\theta}$ be the set of all ODE parameters. We assume that some prior distribution $p_\lambda(\boldsymbol{\theta})$ generates the parameters. Similar to the (intractable) problem of inferring the latent cell time and state variables, we can use a variational approximation of the posterior $q_\phi(\boldsymbol{\theta})$. The marginal likelihood of the input features can be bounded by

$$\log_\lambda(\mathbf{X}) \geq \mathbb{E}_{q_\phi(\theta)}\left[\log\left(p_\theta(\mathbf{X})\right)\right] - KL\left(q_\phi(\theta) \,\|\, p_\lambda(\theta)\right)$$

$$\geq \mathbb{E}_{q_\phi(\theta)}\left[ELBO(\mathbf{X};\theta)\right] - KL\left(q_\phi(\phi) \,\|\, p_\lambda(\theta)\right)$$

$$(11)$$

For the prior $p_\lambda(\boldsymbol{\theta})$, we choose a factorized log-normal distribution, i.e., each rate parameter $\alpha$, $\beta$ or $\gamma$ is a random variable drawn from a log-normal distribution. By default, the logarithm of rate parameters has a mean of zero and a standard deviation of 1 for $\alpha$ and 0.5 for $\beta$ and $\gamma$. The posterior mean is initialized using the same method described in the previous paragraph. Again, we can apply the reparameterization trick to take samples of $\boldsymbol{\theta}$ and estimate the expectation of ELBO with a sample mean. The KL divergence between $q_\phi(\boldsymbol{\theta})$ and $p_\lambda(\boldsymbol{\theta})$ can be viewed as a regularization of the ODE parameters.

This statistical approach also has a nice intuition in our case: it prevents the "vanishing velocity" problem. If we don't regularize $(\alpha, \beta, \gamma)$, optimizing the evidence lower bound may lead to large rate parameters, so that $e^{-\beta\tau} \approx e^{-\gamma\tau} \approx 1$. This in turn causes the velocity to vanish so that each cell will be approximately at the steady state, with $u \approx \frac{\rho\alpha}{\beta}$ and $s \approx \frac{\rho\alpha}{\gamma}$. Because $\rho$ is given by a neural network $g$, we can achieve good reconstruction if $g$ is expressive enough, but the velocity will be close to zero. This issue is resolved by placing a prior distribution on the rates and including them in the variational approximation. This has the effect of regularizing the posterior distribution of the rates toward their prior, avoiding the vanishing velocity problem.

**5.4.7 Interpreting rate parameters.** Although the model accepts any notion of time and rates, the units can be converted to match the actual units of cell developmental time, usually in days. Suppose the cell time from the model ranges from $t_0$ to $t_1$, in a hypothetical time unit, and we have prior knowledge about the entire duration of $m$ days. From the mathematical property of our ODE system, we know that scaling time by $k$ and rate parameters $(\alpha, \beta, \gamma)$ by $\frac{1}{k}$ results in the same $u$ and $s$. Thus, we can perform unit analysis to convert the rates into units of minutes as follows:

$$\alpha \left(\frac{\text{molecule}}{\text{time unit}}\right) = \alpha \cdot \frac{t_1 - t_0}{m} \left(\frac{\text{molecule}}{\text{day}}\right) = \frac{1440(t_1 - t_0)}{m} \alpha \left(\frac{\text{molecule}}{\text{minute}}\right) \tag{12}$$

Similarly, $\beta \left(\frac{1}{\text{time unit}}\right) = \frac{1440(t_1 - t_0)}{m} \beta \left(\frac{1}{\text{minute}}\right)$ and $\gamma \left(\frac{1}{\text{time unit}}\right) = \frac{1440(t_1 - t_0)}{m} \gamma \left(\frac{1}{\text{minute}}\right)$

We can then scale the rate parameters to account for the fact that scRNA-seq captures only a fraction of the transcripts in a cell. A reasonable estimate is about 360000 transcripts in a typical mammalian cell (assuming 10 pg total RNA per cell, 1–2% mRNA, average transcript length of 2kb). Therefore, we can scale $\alpha$ by $\frac{360000}{x_{total}}$, where $x_{total}$ is the median total mRNA count number. We analyzed rate parameters learned from our model, converted the units and computed the histograms (S15 Fig). For $\alpha$, we analyzed the peak transcription rate by considering the case of highest transcription ($\rho = 1$). For $\beta$ and $\gamma$, we multiplied them by $u_{top}$ and $s_{top}$ respectively to obtain the same units as $\alpha$. Here, $u_{top}$ and $s_{top}$ are the 95-percentile $u$ and $s$ values, representing cells with high expression levels.

## 5.5 Solving the chemical master equation with discrete VeloVAE

The biological process of RNA splicing is a chemical reaction network described by the chemical master equation. It considers the change of molecule numbers, which are discrete quantities, as a stochastic process. As Bergen et al. pointed out [10], the dynamical model of RNA velocity is justified by considering the first moment of a stochastic process.

Here, we close the gap between the chemical master equation and its moment equation by introducing the discrete VeloVAE model. We start with chemical equations of all reactions. Consider a gene with transcription, splicing and degradation. For any gene $g$, the reactions are described by the following chemical equations:

$$\emptyset \xrightarrow{\alpha\rho} U_g$$
$$U_g \xrightarrow{\beta} S_g$$
$$S_g \xrightarrow{\gamma} \emptyset \tag{13}$$

Suppose we have a total of $G$ genes. Then, there are $3 \times G$ reactions and $2 \times G$ molecules in total. Since all reactions are monomolecular, the system has a closed form solution if the initial condition is a product Poisson distribution. This is formally described as the following theorem from a previous work [24]:

**Definition 4.** *A product Poisson distribution with mean* $\boldsymbol{\lambda} = [\lambda_1, \ldots, \lambda_d]$ *is a discrete distribution with a support of* $\mathbb{N}^d$ *and PMF of the following form:*

$$P(x; \lambda) = \prod_{i=1}^{d} \frac{e^{-\lambda_i} \lambda_i^k}{k!}$$

(14)

**Theorem 1. (Jahnke, 2007).** *Let P(t,**x**) be the pmf of a feature vector (mRNA counts) at time t. If P(0, **x**) is a product Poisson distribution, then, P(t, **x**) at any time t > 0 is a product Poisson distribution whose mean is given by the following ODE:*

$$\frac{d\lambda}{dt} = A\lambda + b$$

(15)

Here the matrix $A$ contains splicing and degradation rates while $b$ is a vector of transcription rates. An exact solution of a single-gene system, as a special case, is presented in another work [50]. The discrete VeloVAE model is based on the Poisson initial condition assumption. By theorem 1, solving the chemical master equation is equivalent to solving an ODE describing the Poisson mean. In the discrete VeloVAE model, we again use the variational mixture of ODEs as a cell-specific ODE model for the Poisson mean. The generative process is the same as the continuous model except for the last few steps:

$$\begin{aligned}
\boldsymbol{x_{ind}} &\sim \text{Poisson}\left(l \cdot F_{ind}(t; \theta), \Sigma_{\mathbf{r}}\right) \\
\boldsymbol{x_{rep}} &\sim \text{Poisson}\left(l \cdot F_{rep}(t; \theta), \Sigma_{\mathbf{r}}\right) \\
\boldsymbol{x} &= I_{\{w=1\}} \odot \boldsymbol{x_{ind}} + I_{\{w=2\}} \odot \boldsymbol{x_{rep}}
\end{aligned}$$

(16)

Here, we still have a mixture of basis functions for each gene. However, the mixture components are Poisson distributions instead of Gaussian distributions. Besides, we have an additional cell-wise library size scaling factor $l$ to account for differences in count numbers resulting from different sequencing depths.

Because count numbers are drawn from a Poisson distribution, we define the RNA velocity of a discrete model as $v = \frac{d\lambda_s}{dt}$, where $\lambda_s$ is the Poisson rate as well as the mean. This is a natural extension from the continuous model.

### 5.6 Branching ODE

**5.6.1 Model description.** A variational mixture of ODEs provides sufficient flexibility to account for complex kinetics, but only describes such kinetics at the cell level. In order to distill qualitative knowledge about gene expression kinetics and reveal cell-type relations, we propose a new ODE model called branching ODE.

The fundamental assumptions we make about branching ODE include:

1. Each cell belongs to exactly one of the cell types $y_1, \ldots, y_k$.

2. At least one of the cell types is a stem cell type. Each non-stem-cell type is a descendant of exactly one other cell type. (Note that we could relax this assumption to allow multiple progenitors, but we choose not to purely for simplicity.) Each cell type has an initial time $t_0$ when it emerges in the differentiation process.

3. Cells of the same type, $y$, share identical transcription, splicing and degradation rates $(\alpha_y, \beta_y, \gamma_y)$.

With these assumptions, we can summarize differentiation with directed cell type transition graph $G = (V = \{v_1, \ldots, v_k\}, E)$, called a transition graph. Each cell type corresponds to a vertex in $G$, and each edge $(u,v)$ represents the relation of $u$ differentiating into $v$. By our second assumption, $G$ is composed of one or multiple trees. The kinetic functions retain the same analytical form, except for type-specific rate parameters and initial conditions. Furthermore, the equations are defined recursively because the initial condition of any non-stem-cell type depends on its progenitor cell type.

$$u(t, y) = u_0(y)e^{-\beta_y\tau} + \frac{\alpha_y}{\beta_y}\left(1 - e^{-\beta(y)\tau}\right) \tag{17}$$

$$s(t, y) = s_0(y)e^{-\gamma(y)\tau} + \frac{\alpha_y}{\gamma_y}\left(1 - e^{-\gamma_y\tau}\right) + \frac{\alpha_y - \beta_y u_0(y)}{\gamma_y - \beta_y}\left(e^{-\beta_y\tau} - e^{-\gamma_y\tau}\right) \tag{18}$$

$$u_0(y) = \begin{cases} u_{init} & y \text{ is a stem cell type} \\ u(t_0(y), \text{par}(y)) & \text{otherwise} \end{cases} \tag{19}$$

$$s_0(y) = \begin{cases} s_{init} & y \text{ is a stem cell type} \\ s(t_0(y), \text{par}(y)) & \text{otherwise} \end{cases} \tag{20}$$

Here, par($\cdot$) refers to the parent vertex in the transition graph. Note that $\tau$ is redefined as $\tau := t - t_0(y)$ for each cell type $y$, i.e., $\tau$ is the time duration starting from the initial time of each cell type.

**5.6.2 Inferring the transition graph.** A challenging problem for applying branching ODE is the absence of the transition graph. In many cases, we do have some prior knowledge of the transition graph, but it is more desirable to simply infer the graph directly from the data when possible. Therefore, we apply a computational method to infer the transition graph based on cell time and states. Because the transition graph is a collection of arborescences, i.e., rooted trees in a directed graph, we can solve the problem with simple graph algorithms.

First, we partition the cells into distinct cluster(s). In particular, we perform Leiden clustering with a low resolution on the UMAP coordinates of the data.

Next, we build a complete subgraph in each partition. Let $\mathcal{I}$ be the set of all cells and $\mathcal{I}(y)$ be the set of all cells of type $y$ in a partition. For each cell $i \in \mathcal{I}$ with time $t_i$, we take a time window $[t_i - \delta_1, t_i - \delta_2]$ and find k nearest neighbors based on cell state $\mathbf{c}$. Denote $\mathcal{J}_i$ as the set of neighbors of $i$. Then, for any two cell types $y$ and $z$, the empirical transition probability from $y$ to $z$ is defined as

$$P(y, z) = \frac{\sum_{i \in \mathcal{I}(z)} |\mathcal{J}_i \cap \mathcal{I}(y)|}{\sum_{i \in \mathcal{I}(z)} |\mathcal{J}_i|} \tag{21}$$

In other words, we group cells into the cell types and count the number of transitions from any cell type to any other type.

Finally, we apply Edmond's algorithm [51] to find the maximum spanning arborescence in each partition. The earliest cell type is choosen to be the root. The algorithm starts by picking the parent $y$ of each non-root vertex $z$ greedily, i.e., $y = \arg\max_v P(v, z)$. Next, it checks loops, collapses loops into super-vertices and is recursively applied to the new graph until no loop exists. Finally, it breaks the loops after each level of recursion.

**5.6.3 Training the branching ODE.** We assume that the cell time has already been inferred from VeloVAE. Thus, the branching ODE is a regression model with an analytical form shown in equation (17) and (18). To train the model, we find the cell-type-specific rate parameters $\theta$ that maximize the Gaussian likelihood, which is equivalent to minimizing a Mahalanobis distance:

$$\min_\theta \sum_{i=1}^{n} (\hat{\mathbf{x}}(t; \theta) - \mathbf{x})^T \Sigma (\hat{\mathbf{x}}(t; \theta) - \mathbf{x}) \tag{22}$$

Here, $\hat{\mathbf{x}}$ is predicted expression level. We train using mini-batch stochastic gradient descent with the ADAM optimizer.

## 5.7 Data preprocessing

We loaded all datasets from AnnData (h5ad) format. We first used scanpy [52] to select highly variable genes, normalize and scale the gene expression counts. Next, we followed the scVelo preprocessing pipeline by performing principal

component analysis on the normalized and scaled expression data, then smoothing the unspliced and spliced expression levels among $k$-nearest neighbors identified from the principal components.

## 5.8 Evaluation metrics

### 5.8.1 Cross-Boundary Direction Correctness (CBDir).

In a previous work [17], the authors quantitatively measured the accuracy of velocity flow on any 2D visualization. Denote $\mathcal{C}$ as the set of all cell types. For any two cell types $A, B \in \mathcal{C}$, we define $(A,B)$ to be a transition pair if and only if $B$ is a descendant cell type of $A$ in a cell developmental process. The CBDir metric takes a set of know transition pairs $\mathcal{T} = \{(A, B) : A, B \in \mathcal{C}\}$ as input and computes an average directional accuracy via cosine similarity. For any cell in $c \in A$ with transition pair $(A,B)$, its CBDir is defined as:

$$CBDir(c) := \frac{1}{|N(c) \cap C_B|} \sum_{c' \in N(c) \cap C_B} \frac{v_c^T(x_{c'} - x_c)}{\|v_c\| \|x_{c'} - x_c\|} \tag{23}$$

Here, $v_c$ is the projection of RNA velocity to a low-dimensional space, $x_c$ is the coordinates of $c$ in the same space, and $N(c)$ is the set of all neighbors of $c$ in a KNN graph built from gene expression similarity. The overall CBDir of all known transition pairs is defined as

$$CBDir := \frac{1}{|\mathcal{T}|} \sum_{(A,B) \in \mathcal{T}} \frac{1}{|C_A|} \sum_{i \in C_A} CBDir(c) \tag{24}$$

### 5.8.2 Generalized CBDir.

The CBDir metric is the first one directly measuring the correctness of velocity flow. It connects visual perception of velocity flow to a quantity. We think it beneficial to extend it for the following reasons:

1. The velocity in CBDir is only a projection to a low-dimensional space. Hence, it's not a direct measure of velocity at a gene expression level.

2. Only the direct neighbors in the KNN graph are considered as targets, i.e., future transcriptomic states. In fact, such assumption might not hold due to stochastic noise in the dataset. Directly connected neighbors on a KNN graph are too similar at the transcriptomic level to be considered as a future direction.

3. CBDir doesn't take the ordering of cells into account. Time order is important because any real-world system must proceed with increasing time.

4. It remains unclear whether velocity flow from one cell type to another is statistically significant.

Due to all the limitations described above, we devised k-step CBDir (KCBDir) metric. Given a cell $c \in C_A$ with transition pair $(A,B)$, KCBDir is defined as follows:

$$KCBDir(c) := \frac{1}{|N_k(c) \cap C_B|} \sum_{c' \in N_k(c) \cap C_B} (-1)^b \frac{|v_c^T(x_{c'} - x_c)|}{\|v_c\| \|x_{c'} - x_c\|} \tag{25}$$

where $b := I_{\{t_{c'} > t_c \, \wedge \, v_c^T(x_{c'} - x_c) > 0\}}$. Here, we replace direct neighbors with k-step neighbors, $N_k(c)$, on a KNN graph. Besides, $x_c$ and $v_c$ can be from either a low-dimensional embedding or the spliced counts. We further defined the generalized CBDir (GCBDir) by averaging over multiple different step sizes and removing an offset induced by taking a random walk on a KNN graph:

$$GCBDir := KCBDir(c; N_k(c)) - KCBDir(c; R_k(c)) \tag{26}$$

Here, $KCBDir(c;N_k(c))$ is the KCBDir between $c$ and its k-step neighbors from its descendant cell type, and $R_k(c)$ is the set of cells reached by randomly traversing a KNN graph starting from $c$.

### 5.9 Transcription rate comparison

For the Scifate2 dataset [31], metabolic labeling counts were measured over a 2-hour period. We defined the ground-truth transcription rate by dividing the measured labeled counts by the labeling duration (2 hours). To ensure comparability with the transcription rates predicted by continuous models (VeloVAE, DeepVelo, and cellDancer), the ground-truth transcription rates were smoothed using library-size normalization followed by K-nearest neighbor (KNN) averaging. To obtain a biologically meaningful comparison, both the real and predicted transcription rates are multiplied by 100,000 to generate transcripts-per-million-like (TPM-like) measurements. We evaluate model performance using two metrics: mean absolute error (MAE) and the Pearson correlation coefficient (PCC). MAE is computed using the TPM-like normalized transcription rates described above to compare the absolute differences between predicted and ground-truth values. For correlation analysis, the PCC is calculated between the KNN-averaged ground-truth transcription rates and the model-predicted transcription rates per gene across cells.

### 5.10 Time and resource consumption

We evaluate the time and computational resources required for preprocessing, model initialization, and mini-batch training on the Pancreas dataset. The preprocessing step took approximately 8 seconds to select 2,000 genes, followed by around 30 seconds for model initialization. The mini-batch training phase required roughly 5 minutes to complete. All computations were performed on an NVIDIA A40 GPU with 40 GB of RAM.

## Supporting information

**S1 Fig. Comparison of other performance metrics.** We compared VeloVAE with all other methods in terms of Generalized CBDir on both gene expression space **(A)** and 2D low-dimensional embedding **(B)**, velocity accuracy on both gene expression space **(C)** and 2D low-dimensional embedding **(D)** and time accuracy. All metrics are computed based on prior knowledge of cell development in each tissue. Note that we only compared the genes which scVelo fitted, i.e., velocity genes, as there are genes scVelo didn't fit or fitted with zero likelihood. We used tsne for the human bone marrow dataset and umap for all the other datasets as the 2D low-dimensional embedding.
(PNG)

**S2 Fig. Histogram of scVelo and VeloVI time correlation.** We computed the the Spearman correlation between scVelo **(A)** and VeloVI **(B)** locally fitted (gene-specific) time and their global latent time respective for twelve benchmarking datasets. This gives a correlation value for each fitted gene. The figures show box plots of these correlations.
(PNG)

**S3 Fig. Inferred time variance.** Each panel shows a UMAP or t-SNE plot colored by the coefficient of variation of the inferred cell time. The 7 panels correspond to pancreas **(A)**, erythroid **(B)**, subsampled mouse brain**(C)**, iPSC **(D)**, human bone marrow **(E)**, full mouse brain **(F)** and mouse embryo **(G)**.
(PNG)

**S4 Fig. Inferred cell state uncertainty.** Each panel shows a UMAP or t-SNE plot colored by the multi-variate coefficient of variation of the cell state. Cell state is a continuous identification of cell type, so conceptually it should have high uncertainty in progenitor cell types, as cell fate is undetermined. Our results verified this intuition as well as the capability of learning meaningful representations using VeloVAE. The 7 panels correspond to pancreas **(A)**, erythroid **(B)**, subsampled mouse brain **(C)**, iPSC **(D)**, human bone marrow **(E)**, full mouse brain **(F)** and mouse embryo **(G)**.
(PNG)

**S5 Fig. Cell state-time plot.** 2D UMAP embeddings of the latent cell state **c** plotted versus inferred latent time. Each plot has the 2D UMAP embedding as the x-y plane and cell time as the z axis. The plots can be interpreted as time evolution of the cell state space. The seven panels correspond to pancreas **(A)**, erythroid **(B)**, subsampled mouse brain **(C)**, iPSC **(D)**, human bone marrow **(E)**, full mouse brain **(F)** and mouse embryo **(G)**.
(PNG)

**S6 Fig. 3D velocity plot.** 3D plots computed as described in S3 Fig, with velocity vectors added. The x-y plane is either UMAP (A-E) or t-SNE (F-G) coordinates. The 3D embedding is computed by averaging displacement vectors towards k neareast neighbors in the immediate future, i.e., arrows point upwards along the z axis. The seven panels correspond to pancreas **(A)**, erythroid **(b)**, subsampled mouse brain**(C)**, iPSC **(D)**, human bone marrow **(E)**, full mouse brain **(F)** and mouse embryo **(G)**.
(PNG)

**S7 Fig. Distribution of per-cell transcription rate correlations on the Scifate2 dataset.** For each cell, the PCC is computed between predicted and ground-truth transcription rates across all genes. The histogram shows the mean PCC per cell for VeloVAE, DeepVelo, and cellDancer.
(PNG)

**S8 Fig. VeloVAE Resolves Cellular Dynamics of Multiple Lineages Across the Entire Developing Mouse Brain. (A)** t-SNE plots colored by published cell types overlaid with velocity inferred by VeloVAE. **(B)** Cell state-time plot for developing mouse brain dataset. **(C)** Examples of individual genes fit by VeloVAE. Gene fits are shown for both *u* and *s* values, with inferred latent time on the x-axis. Only fitted values are shown for VeloVAE, because the VeloVAE fit is a point cloud (rather than a line) that would completely cover the observed data. Points are colored by published cluster assignments, with the same colors as in the corresponding UMAP plots. Note that the x-axis values are different for scVelo and VeloVAE because they infer different latent times. The arrow length and direction indicate the velocity inferred for each cell. **(D)** t-SNE plots colored by true capture time and latent time inferred by VeloVAE. **(E)-(F)** t-SNE plots colored by the cell state uncertainty **(E)** and cell time variance **(F)** from VeloVAE.
(PNG)

**S9 Fig. Latent time of major trajectories from the mouse embryo.** We compare the capture time, monocle 3 pseudotime and VeloVAE latent time by showing the UMAP plots colored by latent time of ten major trajectories in the embryo dataset: endothelial**(A)**, epithelial **(B)**, haematopoiesis **(C)**, hepatic **(D)**, mesenchymal **(E)**, melanocyte **(F)**, PNS glia **(G)**, PNS neuron **(H)** and neural tube and notochord **(I)** trajectories.
(PNG)

**S10 Fig. Velocity stream plots for major trajectories from the mouse embryo.** We plotted the velocity embedding on UMAP coordinates of ten major trajectories in the embryo dataset: endothelial **(A)**, epithelial **(B)**, haematopoiesis **(C)**, hepatic **(D)**, mesenchymal **(E)**, melanocyte **(F)**, PNS glia**(G)**, PNS neuron **(H)** and neural tube and notochord **(I)** trajectories.
(PNG)

**S11 Fig. Cell state uncertainty in major mouse embryo trajectories.** Each panel shows a UMAP plot colored by the multi-variate coefficient of variation of the cell state. The 10 panels correspond to endothelial **(A)**, epithelial **(B)**, haematopoiesis **(C)**, hepatic **(D)**, mesenchymal **(E)**, melanocyte **(F)**, PNS glia **(G)**, PNS neuron **(H)** and neural tube and notochord**(I)** trajectories in the mouse embryo dataset.
(PNG)

**S12 Fig. Cell time variance in major mouse embryo trajectories.** Each panel shows a UMAP plot colored by the coefficient of variation of the cell time. The 10 panels correspond to endothelial **(A)**, epithelial **(B)**, haematopoiesis **(C)**, hepatic **(D)**, mesenchymal **(E)**, melanocyte **(F)**, PNS glia **(G)**, PNS neuron **(H)** and neural tube and notochord **(I)** trajectories in the mouse embryo dataset. (PNG)

**S13 Fig. Cell state-time plot for additional benchmarked datasets.** We computed 2D UMAP embeddings of the latent cell state **c** and plot it versus inferred latent time. Following the setting of the original work, we used cosine distance and 15 neighbors. The 10 panels correspond to endothelial **(A)**, epithelial **(B)**, haematopoiesis **(C)**, hepatic **(D)**, mesenchymal **(E)**, melanocyte **(F)**, PNS glia**(G)**, PNS neuron **(H)** and neural tube and notochord **(I)** trajectories in the mouse embryo dataset. (PNG)

**S14 Fig. 3D velocity plot for additional benchmarked datasets.** 3D velocity plot computed as described in S6 Fig. The 10 panels correspond to endothelial **(A)**, epithelial **(B)**, haematopoiesis **(C)**, hepatic **(D)**, mesenchymal **(E)**, melanocyte **(F)**, PNS glia **(G)**, PNS neuron **(H)** and neural tube and notochord **(I)** trajectories in the mouse embryo dataset. (PNG)

**S15 Fig. Rate parameter histograms.** Box and histogram plots showing the overall range of transcription, splicing and degradation rates estimated by different methods. For VeloVAE, transcription rates are reported as $\rho = 1$. For splicing and degradation, we report $\beta u$ and $\gamma s$ to match the unit of $\alpha$ (mRNA / minute). Here, $u$ and $s$ are chosen to be half of the 95th-percentile count number. The first four panelsT correspond to pancreas **(A)**, erythroid **(B)**, iPSC **(C)**, subsampled mouse brain **(D)**, full mouse brain **(E)**, and mouse embryo **(F)** datasets. (PNG)

**S1 Table. Default Hyperparameters of VeloVAE.** We trained our model with minimal change to the default hyperparameters. The only changes we made include (1) setting early stop to 9 and train ton to False for the erythroid dataset (2) increasing the batch size to 2048 and n neighbors to 30 for the mouse embryo dataset. (DOCX)

**S2 Table. Default Hyperparameters of Branching ODE.** (DOCX)**S3 Table. Variable Hyperparameters across Datasets.** We utilized the capture time in certain datasets and reversed gene mode assignment to avoid bad initialization. (DOCX)

## Acknowledgments

We would like to thank Jian Shu, Koseki Kobayashi-Kirschvink and Chengxiang Qiu for help with preprocessing the iPSC reprogramming and whole mouse embryo datasets.

## Author contributions

**Conceptualization:** Yichen Gu, David Blaauw, Joshua D. Welch.

**Data curation:** Yuxuan Song.

**Formal analysis:** Yichen Gu, Yuxuan Song, Joshua D. Welch.

**Funding acquisition:** Joshua D. Welch.

**Methodology:** Yichen Gu.

**Project administration:** Joshua D. Welch.

**Software:** Yichen Gu, Yuxuan Song.

**Supervision:** David Blaauw, Joshua D. Welch.

**Visualization:** Yichen Gu, Yuxuan Song.

**Writing – original draft:** Yichen Gu, Yuxuan Song, Joshua D. Welch.

**Writing – review & editing:** Yuxuan Song, Joshua D. Welch.

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
