## [Decision Letter · Decision Letter 0]

30 May 2025

PCOMPBIOL-D-25-00670

Bayesian Inference of RNA Velocity Incorporating Time Points, Lineage Bifurcations, and Count Data

PLOS Computational Biology

Dear Dr. Welch,

Thank you for submitting your manuscript to PLOS Computational Biology. After careful consideration, we feel that it has merit but does not fully meet PLOS Computational Biology's publication criteria as it currently stands. Therefore, we invite you to submit a revised version of the manuscript that addresses the points raised during the review process.

Please submit your revised manuscript within 30 days Jul 30 2025 11:59PM. If you will need more time than this to complete your revisions, please reply to this message or contact the journal office at ploscompbiol@plos.org. Please include the following items when submitting your revised manuscript:

We look forward to receiving your revised manuscript.

Kind regards,

Wei Li, Ph.D.

Academic Editor

PLOS Computational Biology

Ilya Ioshikhes

Section Editor

PLOS Computational Biology

**Journal Requirements:**

At this stage, the following Authors/Authors require contributions: Yichen Gu, David Blaauw, and Joshua Welch. Please ensure that the full contributions of each author are acknowledged in the "Add/Edit/Remove Authors" section of our submission form.

5) We notice that your supplementary Figures, and Tables are included in the manuscript file. Please remove them and upload them with the file type 'Supporting Information'. Please ensure that each Supporting Information file has a legend listed in the manuscript after the references list.

6) Please amend your detailed Financial Disclosure statement. This is published with the article. It must therefore be completed in full sentences and contain the exact wording you wish to be published. State what role the funders took in the study. If the funders had no role in your study, please state: "The funders had no role in study design, data collection and analysis, decision to publish, or preparation of the manuscript.".

**Reviewers' comments:**

Reviewer's Responses to Questions

**Comments to the Authors:**

Reviewer #1: I have reviewed this paper in 2022 for another journal. The authors have extensively revised the manuscript by adding more benchmarking methods on multiple datasets. I appreciated the authors' persistence in pushing it as a useful tool with good reference benchmarking.

As all my major concerns have already been addressed. Here are a few minor things that the authors may consider:

1. Since over the years, there are multiple types of RNA velocity methods proposed, and the authors have included the major ones in the introduction, I would suggest the authors make a table to summarize the features (possibly pros & cons) of each method, including VeloVAE itself.

2. Since the model can leverage the experimental time as prior, I would suggest further highlighting this advantage, even transferring it to non-time series data. This setting can be particularly useful, considering the large time-series atlas are more accessible in multiple systems.

3. The CBDir was originally introduced by VeloAE (PNAS 2021).

Reviewer #2:

1.Summary of the research and overall impression

Cell fate determination is the core question for many developmental biology processes. To model the cell fate complexity, trajectory inference with pseudotime and RNA velocity in single cell RNA-seq has been developed for years. RNA velocity as a popular concept, there are plenty of methods released after Gioele’s initial work. Though these methods have promising application in different biological context, many problems existed in the published methods that limits their usage in complex biology question. The authors presented two models, discrete VeloVAE and Branching ODE, to address one of the key limitations for complex multi-branch (multi-furcation) modeling.

Their key strength is to introduce a mixed of multiple generative processes with variable transcriptional rate across cells or branched cell type specific ODEs to directly model fate from read count data. This divide and conquer strategy not only improves the overall interpretation but also makes the model more robust for different induction/repression dominant genes. Their model is also the only model that explicitly estimate the prior for the physical time points, thus the latent time has been conferred with the real meaning than the other models thus have the potential for longitudinal single cell application.

The main disadvantage is a lacking true quantitative approach to validate their approach such as those profiling in single cell lineage tracing experiments for cell fate, and those profiling in metabolic labelling single cell experiment for kinetic rate. Another main disadvantage of the study is the lacking documentation about how to reproduce their computational experiments.

Overall, this manuscript describes a solid model that will bring values for the RNA velocity extensions, and the benchmarking results are convincingly good that VeloVAE is among the one of the best performers and resolved some of the previous modeling errors for gene phase portraits and latent time.

2.Issues

Major issues

a.The authors have written sufficient details of their velocity method, however, there is missing section in Methods that how to make reproducible experiments in their benchmarking, for example, how many top variable genes selected, which method used to generate vector field from the transition matrix, how many epochs run for a given method. This is critically important for other deep learning/Bayesian based approach, since their user option selection has large influenced on the model performance. A Reproducibility text section for all the tools is needed, also the codes for the Reproducibility of their method and other methods should be provided somewhere.

b.So far, the quantitative metrics to evaluate the involved velocity methods in this manuscript are more like qualitative based on the (k-step) CBDir or spearman correlation with discrete time points with prior knowledges. However, the real ground truth data is worth applied with the VeloVAE and similar methods in new technologies such as scGestalt, Celltagging, CARLIN, or LARRY. These profiling can provide near single cell level labels for the developmental orders based on their clonal barcoding. The same question also holds for the kinetic rate benchmarking, the authors could consider those released single cell metabolic data from dynamo to benchmark the errors of cell specific transcription rate inference.

c.Pyro-Velocity is the most similar method to VeloVAE in terms of raw read counts modeling and inexplicit gene specific ODE but with different way, thus it is worth more in-depth comparison than the other methods. Pyro-Velocity model II used an offset variable that allows the selection of the genes that belongs to fully induction or fully repression phase. First, it is not clear which Pyro-Velocity model has been used in the manuscript given the poor fitting performance, it might be model 1 then it is not a fair comparison. Second, other comparison such as gene phase portraits needs to be added based on the model 2.

d.In Figure 2a, how many time points are here? If there are only two time points, since the capture time are limited to a few discrete points, rank-based spearman correlation may not be a suitable metric for evaluation of latent time. Boxplots grouped by these time points and compared latent time with pairwise statistical test by ordering with time on y axis might be more suitable.

e.Probabilistic Bayesian model is vulnerable to the initialization and random shuffling of mini batches. We noticed different initialization in Table S3 has been used for different datasets. In general, would the authors describe the decision of hyperparameters search strategy in their experiments?

f.The method labels on the figures are a bit confusing, what are the differences between VeloVAE, Discrete VeloVAE, Discrete FullVB, and FullVB on Fig 2 a-c? Is FullVB representing the Full variational Bayesian? Which one is the main VeloVAE model applied to the rest of the figures?

Minor Issues

a.Fig S3-S5 are not mentioned in the text at all, please add brief description.

b.In Fig.3, it is a crowded figure with the same colors meaning different cell types across datasets, it might be better to layout the panels coming from the same dataset together, e.g., d with i.j together.

c.scVelo can actually perform out-of-sample prediction if we plugin in the gene wise parameters with the inverse function of gene expression to tau.

d.In Fig.S1b, what 2D embeddings are used here? UMAP or PCA?

e.In Fig.S2, is a panel for the scvelo and b for veloVI?

f.Overall runtime for processed steps (testing induction or full phase genes, selecting genes) and modeling steps will be helpful for users’ references.

g.Describe the potentially existing bottleneck for the VeloVAE model, e.g. when the model will fail, and future possible extension in the Discussion section. For example, the complexity of the deep neural network for inferring posterior might not fit the small scale dataset such as the scvelo dentate gyrus data.

h.Please remove the large text gap in the page 4 and 14.

i.Section 2.2 “and and log likelihood” should be “and log likelihood”.

**Have the authors made all data and (if applicable) computational code underlying the findings in their manuscript fully available?**

The PLOS Data policy requires authors to make all data and code underlying the findings described in their manuscript fully available without restriction, with rare exception (please refer to the Data Availability Statement in the manuscript PDF file). The data and code should be provided as part of the manuscript or its supporting information, or deposited to a public repository. For example, in addition to summary statistics, the data points behind means, medians and variance measures should be available. If there are restrictions on publicly sharing data or code —e.g. participant privacy or use of data from a third party—those must be specified.requires authors to make all data and code underlying the findings described in their manuscript fully available without restriction, with rare exception (please refer to the Data Availability Statement in the manuscript PDF file). The data and code should be provided as part of the manuscript or its supporting information, or deposited to a public repository. For example, in addition to summary statistics, the data points behind means, medians and variance measures should be available. If there are restrictions on publicly sharing data or code —e.g. participant privacy or use of data from a third party—those must be specified.

Reviewer #1: None

Reviewer #2: **No:** The VeloVAE codes have been available. However, they haven't released codes for the benchmarking part.The VeloVAE codes have been available. However, they haven't released codes for the benchmarking part.

PLOS authors have the option to publish the peer review history of their article (what does this mean?. If published, this will include your full peer review and any attached files.). If published, this will include your full peer review and any attached files.

**Do you want your identity to be public for this peer review?** For information about this choice, including consent withdrawal, please see our For information about this choice, including consent withdrawal, please see our Privacy Policy ..

Reviewer #1: No

Reviewer #2: No

**Figure resubmission:**
---

## [Decision Letter · Decision Letter 1]

23 Dec 2025

PCOMPBIOL-D-25-00670R1

Bayesian Inference of RNA Velocity Incorporating Time Points, Lineage Bifurcations, and Count Data

PLOS Computational Biology

Dear Dr. Welch,

Thank you for submitting your manuscript to PLOS Computational Biology. After careful consideration, we feel that it has merit but does not fully meet PLOS Computational Biology's publication criteria as it currently stands. Therefore, we invite you to submit a revised version of the manuscript that addresses the points raised during the review process.

We look forward to receiving your revised manuscript.

Kind regards,

Wei Li, Ph.D.

Academic Editor

PLOS Computational Biology

Ilya Ioshikhes

Section Editor

PLOS Computational Biology

**Journal Requirements:**

At this stage, the following Authors/Authors require contributions: Yichen Gu, David Blaauw, and Joshua Welch. Please ensure that the full contributions of each author are acknowledged in the "Add/Edit/Remove Authors" section of our submission form.

2) Please amend your detailed Financial Disclosure statement. This is published with the article. It must therefore be completed in full sentences and contain the exact wording you wish to be published.

State what role the funders took in the study. If the funders had no role in your study, please state: "The funders had no role in study design, data collection and analysis, decision to publish, or preparation of the manuscript.".

**Reviewers' comments:**

Reviewer's Responses to Questions

**Comments to the Authors:**

Reviewer #1: All my concerns have been addressed.

Reviewer #2: Summary of comments

The revised manuscript is much improved compared to the original version. I thank the authors for the time and substantial effort they invested in addressing the previous comments. I have one additional minor comment regarding Fig. 4, particularly panels (a) and (e).

Specifically, it is not entirely clear from Fig. 4a how the stream plot supports the statement: "Our stream plot shows that VeloVAE correctly identifies HSCs as the start of differentiation and predicts differentiation into megakaryocytes, platelets, dendritic cells, monocytes, and B cells." In particular, the stream lines appear to originate from a region annotated as CD8 effector cells rather than from HSCs. I suggest explicitly labeling the trajectory paths (e.g., arrows or annotations) from HSCs to the terminal cell states to clarify the inferred differentiation directions and starting point.

For Fig. 4e, the key point is not merely that the latent time contradicts the stream plot, but whether each method reflects the true differentiation trajectories. It would be helpful to explicitly highlight where and how such contradictions arise across the different methods shown, and to clarify which trajectories are considered biologically consistent.

**Have the authors made all data and (if applicable) computational code underlying the findings in their manuscript fully available?**

The PLOS Data policy requires authors to make all data and code underlying the findings described in their manuscript fully available without restriction, with rare exception (please refer to the Data Availability Statement in the manuscript PDF file). The data and code should be provided as part of the manuscript or its supporting information, or deposited to a public repository. For example, in addition to summary statistics, the data points behind means, medians and variance measures should be available. If there are restrictions on publicly sharing data or code —e.g. participant privacy or use of data from a third party—those must be specified.requires authors to make all data and code underlying the findings described in their manuscript fully available without restriction, with rare exception (please refer to the Data Availability Statement in the manuscript PDF file). The data and code should be provided as part of the manuscript or its supporting information, or deposited to a public repository. For example, in addition to summary statistics, the data points behind means, medians and variance measures should be available. If there are restrictions on publicly sharing data or code —e.g. participant privacy or use of data from a third party—those must be specified.

Reviewer #1: Yes

Reviewer #2: Yes

PLOS authors have the option to publish the peer review history of their article (what does this mean?. If published, this will include your full peer review and any attached files.). If published, this will include your full peer review and any attached files.

Reviewer #1: No

Reviewer #2: No

**Figure resubmission:**
---

## [Editor Report · Decision Letter 2]

25 Feb 2026

Dear Dr. Welch,

We are pleased to inform you that your manuscript 'Bayesian Inference of RNA Velocity Incorporating Time Points, Lineage Bifurcations, and Count Data' has been provisionally accepted for publication in PLOS Computational Biology.

Best regards,

Wei Li, Ph.D.

Academic Editor

PLOS Computational Biology

Ilya Ioshikhes

Section Editor

PLOS Computational Biology
